# NEAR-OPTIMAL CORESETS FOR ROBUST CLUSTERING

**Lingxiao Huang**
State Key Laboratory of Novel Software Technology,
Nanjing University
Email: `huanglingxiao1990@126.com`

**Shaofeng H.-C. Jiang**
Peking University
Email: `shaofeng.jiang@pku.edu.cn`

**Jianing Lou**
Peking University
Email: `loujn@pku.edu.cn`

**Xuan Wu**
Huawei TCS Lab
Email: `wu3412790@gmail.com`

## ABSTRACT

We consider robust clustering problems in $\mathbb{R}^d$, specifically $k$-clustering problems (e.g., $k$-MEDIAN and $k$-MEANS) with $m$ *outliers*, where the cost for a given center set $C \subset \mathbb{R}^d$ aggregates the distances from $C$ to all but the furthest $m$ data points, instead of all points as in classical clustering. We focus on the $\epsilon$-coreset for robust clustering, a small proxy of the dataset that preserves the clustering cost within $\epsilon$-relative error for all center sets. Our main result is an $\epsilon$-coreset of size $O(m + \text{poly}(k\epsilon^{-1}))$ that can be constructed in near-linear time. This significantly improves previous results, which either suffers an exponential dependence on $(m + k)$ (Feldman & Schulman, 2012), or has a weaker bi-criteria guarantee (Huang et al., 2018). Furthermore, we show this dependence in $m$ is nearly-optimal, and the fact that it is isolated from other factors may be crucial for dealing with large number of outliers. We construct our coresets by adapting to the outlier setting a recent framework (Braverman et al., 2022) which was designed for capacity-constrained clustering, overcoming a new challenge that the participating terms in the cost, particularly the excluded $m$ outlier points, are dependent on the center set $C$. We validate our coresets on various datasets, and we observe a superior size-accuracy tradeoff compared with popular baselines including uniform sampling and sensitivity sampling. We also achieve a significant speedup of existing approximation algorithms for robust clustering using our coresets.

## 1 INTRODUCTION

We give near-optimal $\epsilon$-coresets for $k$-MEDIAN and $k$-MEANS (and more generally, $(k, z)$-CLUSTERING) with outliers in Euclidean spaces. Clustering is a central task in data analysis, and popular center-based clustering methods, such as $k$-MEDIAN and $k$-MEANS, have been widely applied. In the vanilla version of these clustering problems, given a *center set* of $k$ points $C$, the objective is usually defined by the sum of (squared) distances from each data point to $C$.

This formulation, while quite intuitive and simple to use, has severe robustness issues when dealing with noisy/adversarial data; for instance, an adversary may add few noisy *outlier* points that are far from the center to "fool" the clustering algorithm to wrongly put centers towards those points in order to minimize the cost. Indeed, such robustness issue introduced by outliers has become a major challenge in data science and machine learning, and it attracted extensive algorithmic research on the topic (Charikar et al., 2001; Chen, 2008; Candès et al., 2011; Chawla & Gionis, 2013; Mount et al., 2014; Gupta et al., 2017; Statman et al., 2020; Ding & Wang, 2020). Moreover, similar issues have also been studied from the angle of statistics (Huber & Ronchetti, 2009).

**Robust Clutering** We consider *robust* versions of these clustering problems, particularly a natural and popular variant, called clustering with outliers (Charikar et al., 2001). Specifically, given a dataset $X \subset \mathbb{R}^d$, the $(k, z, m)$-ROBUST CLUSTERING problem is to find a center set $C \subset \mathbb{R}^d$ of $k$

points (repetitions allowed), that minimizes the objective function

$$\text{cost}_z^{(m)}(X, C) := \min_{L \subseteq X : |L| = m} \sum_{x \in X \setminus L} (\text{dist}(x, C))^z. \qquad (1)$$

Here, $L$ denotes the set of *outliers*, dist denotes the Euclidean distance, and $\text{dist}(x, C) := \min_{c \in C} \text{dist}(x, c)$. Intuitively, the outliers capture the furthest points in a cluster which are "not well-clustered" and are most likely to be the noise. Notice that the parameter $z$ captures various (robust) clustering problems, including $(k, m)$-ROBUST MEDIAN (where $z = 1$), $(k, m)$-ROBUST MEANS (where $z = 2$). On the other hand, if the number of outliers $m = 0$ then the robust clustering problem falls back to the non-robust version. The $(k, z, m)$-ROBUST CLUSTERING problem has been widely studied in the literature (Chen, 2008; Gupta et al., 2017; Krishnaswamy et al., 2018; Friggstad et al., 2019; Statman et al., 2020). Moreover, the idea of removing outliers has been also considered in other machine learning tasks, e.g., robust PCA (Bhaskara & Kumar, 2018) and robust regression Rousseeuw & Leroy (1987); Mount et al. (2014).

**Computational Challenges** However, the presence of outliers introduces significant computational challenges, and it inspires a series of research to design efficient algorithms for robust clustering. On one hand, approximation algorithms with strict accuracy guarantee has been obtained (Charikar et al., 2001; Chen, 2008; Gupta et al., 2017; Krishnaswamy et al., 2018; Feng et al., 2019; Friggstad et al., 2019; Zhang et al., 2021) but their running time is a high-degree polynomial which is impractical. On the other hand, more scalable algorithms were also proposed (Bhaskara et al., 2019; Deshpande et al., 2020), however, the approximation ratio is worse, and a more severe limitation is that their guarantee usually violates the required number of outliers. Moreover, to the best of our knowledge, we are not aware of works that design algorithms in sublinear models, such as streaming and distributed computing.

**Coresets** In order to tackle the computational challenges, we consider *coresets* for robust clustering. Roughly, an $\epsilon$-coreset is a tiny proxy of the massive input dataset, on which the clustering objective is preserved within $\epsilon$-error for every potential center set. Existing algorithms may benefit a significant speedup if running on top of a coreset, and more importantly, coresets can be used to derive sublinear algorithms, including streaming algorithms (Har-Peled & Mazumdar, 2004), distributed algorithms (Balcan et al., 2013) and dynamic algorithms (Henzinger & Kale, 2020), which are highly useful to deal with massive datasets.

Stemming from Har-Peled & Mazumdar (2004), the study of coresets for the non-robust version of clustering, i.e., $(k, z)$-CLUSTERING, has been very fruitful (Feldman & Langberg, 2011; Feldman et al., 2020; Sohler & Woodruff, 2018; Huang & Vishnoi, 2020; Braverman et al., 2021; Cohen-Addad et al., 2021b; Braverman et al., 2022), and the state-of-the-art coreset achieves a size $\text{poly}(k\epsilon^{-1})$, independent of $d$ and $n$. However, coresets for robust clustering were much less understood. Existing results either suffers an exponential $(k + m)^{k+m}$ factor in the coreset size (Feldman & Schulman, 2012), or needs to violate the required number of outliers (Huang et al., 2018). This gap leads to the following question: can we efficiently construct an $\epsilon$-coreset of size $\text{poly}(m, k, \epsilon^{-1})$ for $(k, z, m)$-ROBUST CLUSTERING (without violating the number of outliers)?

## 1.1 OUR CONTRIBUTIONS

Our main contribution, stated in Theorem 1.1, is a near-optimal coreset for robust clustering, affirmatively answering the above question. In fact, we not only achieve $\text{poly}(m)$, but also linear in $m$ and is isolated from other factors. This can be very useful when the number of outliers $m$ is large.

**Theorem 1.1** (Informal; see Theorem 3.1). *There exists a near-linear time algorithm that given data set $X \subset \mathbb{R}^d$, $z \geq 1$, $\epsilon \in (0, 0.3)$ and integers $k, m \geq 1$, computes an $\epsilon$-coreset of $X$ for $(k, z, m)$-ROBUST CLUSTERING of size $O(m) + 2^{O(z \log z)} \tilde{O}(k^3 \epsilon^{-3z-2})$, with constant probability.*

Our coreset improves over previous results in several aspects. Notably, compared with Feldman & Schulman (2012), our result avoids their exponential $(k + m)^{k+m}$ factor in the coreset size which is likely to be impractical since typical values of $k$ and/or $m$ may be $O(\log n)$. In fact, as observed in our experiments, the value of $m$ can be as large as 1500 in real datasets, so the dependence in Feldman & Schulman (2012) is prohibitively large which leads to an inferior practical performance

(see Section 4). Moreover, our coreset has a strict guarantee for $m$ outliers instead of a bi-criteria guarantee as in (Huang et al., 2018) that needs to allow more or fewer outliers in the objective for the coreset. We also note that our coreset is composable (Remark 3.2). Furthermore, we show that the linear dependence in $m$ is necessary (Theorem 1.2). Hence, combining this with a recent size lower bound of $\Omega(k\epsilon^{-2})$ (Cohen-Addad et al., 2022) for vanilla clustering (i.e., $m = 0$), we conclude that the dependence of every parameter (i.e., $m, k, \epsilon$) is nearly tight.

**Theorem 1.2.** *For every integer $m \geq 1$, there exists a dataset $X \subset \mathbb{R}$ of $n \geq m$ points, such that for every $0 < \epsilon < 0.5$, any $\epsilon$-coreset for $(1, m)$-ROBUST MEDIAN must have size $\Omega(m)$.*

For the lower bound, we observe that when $m = n - 1$, the clustering cost for $(1, m)$-ROBUST MEDIAN reduces to the distance to the nearest-neighbor from the center $c$. This is easily shown to require $\Omega(n) = \Omega(m)$ points in the coreset, in order to achieve any finite approximation. The formal proof can be found in Section H.

**Experiments** We evaluate the empirical performance of our coreset on various datasets (in Section 4). We validate the size-accuracy tradeoff of our coreset compared with popular coreset construction methods, particularly uniform sampling (which is a natural heuristic) and sensitivity sampling (Feldman & Schulman, 2012), and we observe that our coreset consistently outperforms these baselines in accuracy by a significant margin for every experimented coreset size ranging from 500 to 5000. We also run existing approximation algorithms on top of our coreset, and we achieve about 100x speedup for both a) a Lloyd heuristic adopted to the outlier setting (Chawla & Gionis, 2013) that is seeded by an outlier-version of $k$-MEANS++ (Bhaskara et al., 2019), and b) a natural local search algorithm (Friggstad et al., 2019). These numbers show that our coreset is not only near-optimal in theory, but also demonstrates the potential to be used in practice.

## 1.2 TECHNICAL OVERVIEW

Similar to many previous coreset constructions, we first compute a near-optimal solution, an $(\alpha, \beta, \gamma)$-approximation (see Definition 2.2) $C^* := \{c_i^* \mid i \in [\beta k]\}$, obtained using known approximation algorithms (see the discussion in Section A). Then with respect to $C^*$, we identify the outliers $L^* \subset X$ of $C^*$ and partition the remaining inlier points $X \setminus L^*$ into $|C^*|$ clusters $\{X_i\}_i$.

We start with including $L^*$ into our coreset, and we also include a weighted subset of the remaining inlier points $X \setminus L^*$ by using a method built upon a recent framework Braverman et al. (2022), which was originally designed for clustering with capacity constraints. The step of including $L^*$ in the coreset is natural, since otherwise one may miss the remote outlier points which can incur a huge error; furthermore, the necessity of this step is also justified by our $\Omega(m)$ lower bound (Theorem 1.2). Similar to Braverman et al. (2022), for each cluster $X_i$ among the remaining inliers points $X \setminus L^*$, we identify a subset of $X_i$ that consists of $\text{poly}(k\epsilon^{-1})$ *rings*, and merge the remaining part into $\text{poly}(k\epsilon^{-1})$ *groups* of rings such that each group $G$ has a tiny cost (Theorem 3.3). We use a general strategy that is similar to Braverman et al. (2022) to handle separately the rings (Lemma 3.6) and groups (Lemma 3.7), but the actual details differ significantly due to the presence of outliers.

**Handling Rings** Similar to Braverman et al. (2022), for a ring data subset $R = \text{ring}(c_i^*, r, 2r) \subseteq X_i$, i.e., a subset such that every point is at a similar distance (up to a factor of 2) to the center $c_i^*$, we apply a uniform sampling on it to construct a coreset (with *additive* error, Definition 3.5). In Braverman et al. (2022), for any center set $C \subset \mathbb{R}^d$, the error incurred by uniform sampling is bounded by $\epsilon \cdot \text{cost}_z(R, C)$ which is $\epsilon$ times the total cost *without* outliers from $R$ to $C$ (ignoring some neglectable additive term). However, in the presence of outliers, their error bound $\epsilon \cdot \text{cost}_z(R, C)$ can hardly be charged to $\epsilon \cdot \text{cost}_z^{(m_R)}(R, C)$, where $m_R$ is the number of outliers in $R$ with respect to $C$. This is because $\text{cost}_z^{(m_R)}(R, C)$ can be very small and even close to 0 when $m_R \approx |R|$. Moreover, the number of outliers $m_R$ is not known a priori and can be any number between 0 and $m$. Hence, we provide a stronger guarantee (Lemma 3.6) where we give an alternative upper bound which eventually charges the error to $\epsilon \cdot \text{cost}_z^{(m_R)}(R, C)$ and $\epsilon \cdot \text{opt}$. We use the fact that $\text{cost}_z^{(m_R)}(R, C)$ is "small enough" compared to opt for large $m_R$, while for small $m_R$, we rewrite the robust clustering cost as an integration of ball ranges (Fact F.1) and use a fact that uniform sampling approximately estimates all ball ranges (Lemma F.3). Similar idea of writing the cost as an integration has also been used in previous works, e.g., Huang et al. (2018); Braverman et al. (2022).

**Handling Groups** The main technical difficulty is to handle groups (Lemma 3.7). We still construct a two-point coreset (Definition 3.4) for every group $G \subset X_i$, as in Braverman et al. (2022). To analyze the error of this two-point coreset for an arbitrary center set $C \subset \mathbb{R}^d$, we partition the groups into *colored* and *uncolored* groups *with respect to* $C$ (Lemma G.1) in a way similar to Braverman et al. (2022). Let us call a group $G$ "bad" if the error incurred by the two-point coreset is much larger than $\varepsilon \cdot \text{cost}_z^{(m_G)}(G, C)$, which is $\varepsilon$ times the contribution of $G$. We focus our discussion on bounding the error of bad groups. We first show even with outliers (in Lemma G.1), the error for each bad group is at most $\varepsilon \cdot \text{cost}_z(G, C^*)$. Hence, it remains to bound the number of bad groups. In Braverman et al. (2022), only a colored group can be bad, and the number of them is bounded by $O(k \log \frac{z}{\varepsilon})$. However, in the outlier setting a key difference to Braverman et al. (2022) is that, uncolored groups can also be bad (due to a reason similar to that for rings: $\text{cost}_z^{(m_G)}(G, C)$ may be too small when the number of outliers $m_G$ in $G$ is large), and we call them "special uncolored groups" (22). To bound the number of them, in Lemma G.5 we make a crucial geometric observation: even though special uncolored groups may significantly change for varying $C$'s, an invariant is that they must always be consecutive uncolored groups, due to the way we decompose $X_i$, and apart from the two groups that partially intersect the outliers, every other group within the consecutive sequence consists of outliers only. This key geometric observation implies the number of such groups is $O(1)$ (Lemma G.5), and consequently, the total number of bad groups is at most $O(k \log \frac{z}{\varepsilon})$ with respect to any $C$.

Finally, in addition to the above new steps, we remark that it is also necessary to make these bounds for rings/groups work for all numbers of outliers $0 \leq t \leq m$ "simultaneously", since one does not know in advance how many outliers reside each ring/group due to the arbitrarily chosen $C$.

### 1.3 OTHER RELATED WORKS

**Robust Clustering in $\mathbb{R}^d$** Robust clustering, first proposed by Charikar et al. (2001), has been studied for two decades. For $(k, m)$-ROBUST MEDIAN, Charikar et al. (2001) designed a bi-criteria approximate algorithm with violations on $k$. Chen (2008) first showed a pure constant approximate algorithm, whose approximate ratio was improved to $7.081 + \epsilon$ (Krishnaswamy et al., 2018). When $k = O(1)$, Feng et al. (2019) also proposed a PTAS for $(k, m)$-ROBUST MEDIAN. For $(k, m)$-ROBUST MEANS, Gupta et al. (2017) designed a bi-criteria approximate algorithm with violations on $m$. Krishnaswamy et al. (2018) first proposed a constant approximate algorithm, and the approximate ratio was improved to $6 + \epsilon$ Feng et al. (2019). For general $(k, z, m)$-ROBUST CLUSTERING, Friggstad et al. (2019) achieved an $O(z^z)$ approximate solution with $(1 + \epsilon)k$ centers. Due to the wide applications, scalable algorithms have been designed for $(k, m)$-ROBUST MEANS Bhaskara et al. (2019); Deshpande et al. (2020) besides theoretical study, which may have a worse provable guarantee but are more efficient in practice.

**Coresets for Clustering** There is a large body of work that studies coreset construction for vanilla $(k, z)$-CLUSTERING in $\mathbb{R}^d$ Har-Peled & Mazumdar (2004); Feldman & Langberg (2011); Braverman et al. (2016); Huang et al. (2018); Cohen-Addad et al. (2021b; 2022). The state-of-art result for general $(k, z)$-CLUSTERING is by Cohen-Addad et al. (2022), where the coreset size is $\tilde{O}(z^{O(z)}k\epsilon^{-2} \cdot \min\{k, \epsilon^{-z}\})$. This bound nearly matches a lower bound of $\Omega(k\epsilon^{-2} + k \min\{d, 2^{z/20}\})$ (Cohen-Addad et al., 2022; Huang & Vishnoi, 2020). In addition, coresets for constrained clustering in Euclidean spaces has also been considered, such as capacitated clustering and the tightly related fair clustering (Schmidt et al., 2019; Huang et al., 2019; Braverman et al., 2022), and ordered weighted clustering (Braverman et al., 2019). Going beyond Euclidean spaces, coresets of size $\text{poly}(k\epsilon^{-1})$ were known for $(k, z)$-CLUSTERING in doubling metrics (Huang et al., 2018), shortest-path metrics of graphs with bounded treewidth (Baker et al., 2020) and graphs that exclude a fixed minor (Braverman et al., 2021).

## 2 PRELIMINARIES

**Balls and Rings** For a point $a \in \mathbb{R}^d$, and positive real numbers $r' > r > 0$, define $\text{Ball}(a, r) = \{x \in \mathbb{R}^d, \text{dist}(x, a) \leq r\}$ and $\text{ring}(a, r, r') = \text{Ball}(a, r') \setminus \text{Ball}(a, r)$. For a set of points $A \subset \mathbb{R}^d$, $\text{Balls}(A, r) = \cup_{a \in A} \text{Ball}(a, r)$.

**Weighted Outliers** Since our coreset uses weighted points, we need to define the notion of weighted sets and weighted outliers. We call a set $S$ with an associated weight function $w_S : S \to \mathbb{R}_{\geq 0}$ a *weighted set*. Given two weighted sets $(X, w_X)$ and $(Y, w_Y)$ such that $Y \subseteq X$ and $w_Y(x) \leq w_X(x)$ for any $x \in Y$, let $X - Y$ denote a weighted set $(Z, w_Z)$ such that $w_Z = w_X - w_Y$,[1] and $Z$ is the support of $w_Z$. Moreover, for a weighted set $X$, we denote $\mathcal{L}_X^{(m)}$ as the collection of all possible sets of *weighted outliers* $(Y, w_Y)$ satisfying that $Y \subseteq X$, $\sum_{x \in Y} w_Y(x) = m$ and that $\forall x \in X, w_Y(x) \leq w_X(x)$. In this definition, since $X$ is a weighted point set, we need to pick outliers of *total weight* $m$ in the objective $\mathrm{cost}_z^{(m)}(X, C)$, instead of $m$ distinct points which may have a much larger weights than $m$.

**Weighted Cost Functions** For $m = 0$, we write $\mathrm{cost}_z$ for $\mathrm{cost}_z^{(m)}$. We extend the definition of the cost function to that on a weighted set $X \subset \mathbb{R}^d$. For $m = 0$, we define $\mathrm{cost}_z(X, C) := \sum_{x \in X} w_X(x) \cdot (\mathrm{dist}(x, C))^z$. For general $m \geq 1$, the cost is defined using the notion of weighted outliers and aggregating using the $\mathrm{cost}_z$ function which is the $m = 0$ case.

$$\mathrm{cost}_z^{(m)}(X, C) := \min_{(L, w_l) \in \mathcal{L}_X^{(m)}} \left\{ \mathrm{cost}_z(X - L, C) \right\}.$$

One can check that this definition is a generalization of the unweighted case (1). For a weighted set $X \subset \mathbb{R}^d$, let the optimal solution be $\mathrm{opt}_z^{(m)}(X) := \min_{C \subset \mathbb{R}^d, |C|=k} \mathrm{cost}_z^{(m)}(X, C)$.

**Definition 2.1** (Coreset). Given a point set $X \subset \mathbb{R}^d$ and $\epsilon \in (0, 1)$, an $\epsilon$-coreset for $(k, z, m)$-ROBUST CLUSTERING is a weighted subset $(S, w_S)$ of $X$ such that

$$\forall C \subset \mathbb{R}^d, |C| = k, \qquad \mathrm{cost}_z^{(m)}(S, C) \in (1 \pm \epsilon) \cdot \mathrm{cost}_z^{(m)}(X, C) \tag{2}$$

Even though Definition 2.1 naturally extends the definition of coresets for vanilla clustering (Har-Peled & Mazumdar, 2004; Feldman & Langberg, 2011; Feldman et al., 2020), it is surprising that this exact definition did not seem to appear in the literature. A closely related definition (Huang et al., 2018) considers a relaxed "bi-criteria" (with respect to the number of outliers) guarantee of the cost, i.e., $(1 - \epsilon) \cdot \mathrm{cost}_z^{(1+\beta)m}(S, C) \leq \mathrm{cost}_z^{(m)}(X, C) \leq (1 + \epsilon) \cdot \mathrm{cost}_z^{(1-\beta)m}(S, C)$, for $\beta \in [0, 1)$, and their coreset size depends on $\beta^{-1}$. Another definition was considered in Feldman & Schulman (2012), which considers a more general problem called weighted clustering (so their coreset implies our Definition 2.1). Unfortunately, this generality leads to an exponential-size coreset (in $k, m$).

**Definition 2.2** (($\alpha, \beta, \gamma$)-Approximation). Given a dataset $X \subset \mathbb{R}^d$ and real numbers $\alpha, \beta, \gamma \geq 1$, an $(\alpha, \beta, \gamma)$-approximate solution for $(k, z, m)$-ROBUST CLUSTERING on $X$ is a center set $C^* \subset \mathbb{R}^d$ with $|C^*| \leq \beta k$ such that $\mathrm{cost}_z^{(\gamma m)}(X, C^*) \leq \alpha \cdot \mathrm{opt}_z^{(m)}(X)$.

# 3 CORESETS FOR $(k, z, m)$-ROBUST CLUSTERING

We present our main theorem in Theorem 3.1. As mentioned, the proof of Theorem 3.1 is based on the framework in Braverman et al. (2022), and we review the necessary ingredients in Section 3.1. The statement of our algorithm and the proof of Theorem 3.1 can be found in Section 3.2.

**Theorem 3.1.** *Given input dataset $P \subset \mathbb{R}^d$ with $|P| = n$, integers $k, m \geq 1$, and real number $z \geq 1$ and assume there exists an algorithm that computes an $(\alpha, \beta, \gamma)$-approximation of $P$ for $(k, z, m)$-ROBUST CLUSTERING in time $\mathcal{A}(n, k, d, z)$, then Algorithm 1 uses time $\mathcal{A}(n, k, d, z) + O(nkd)$ to construct a weighted subset $(S, w_S)$ with size $|S| = \gamma m + 2^{O(z \log z)} \cdot \beta \cdot \tilde{O}(k^3 \epsilon^{-3z-2})$, such that with probability at least 0.9, for every integer $0 \leq t \leq m$, $S$ is an $\alpha\epsilon$-coreset $S$ of $P$ for $(k, z, t)$-ROBUST CLUSTERING.*

**Remark 3.2.** By rescaling $\epsilon$ to $\epsilon/\alpha$ in the input of Algorithm 1, we obtain an $\epsilon$-coreset of size $\gamma m + 2^{O(z \log z)} \cdot \alpha^{3z+2} \beta \cdot \tilde{O}(k^3 \epsilon^{-3z-2})$. We discuss how to obtain $(\alpha, \beta, \gamma)$-approximations in Section A. We also note that Theorem 3.1 actually yields an $\epsilon$-coreset for $(k, z, t)$-ROBUST CLUSTERING *simultaneously* for every integer $0 \leq t \leq m$, which implies that our coreset is *composable*. Specifically, if for every integer $0 \leq t \leq m$, $S_X$ is an $\epsilon$-coreset of $X$ for $(k, z, t)$-ROBUST CLUSTERING and $S_Y$ is an $\epsilon$-coreset of $Y$ for $(k, z, t)$-ROBUST CLUSTERING, then for every integer $0 \leq t \leq m$, $S_X \cup S_Y$ is an $\epsilon$-coreset of $X \cup Y$ for $(k, z, t)$-ROBUST CLUSTERING.

---

[1]Here, if $x \notin Y$, we let $w_Y(x) = 0$.

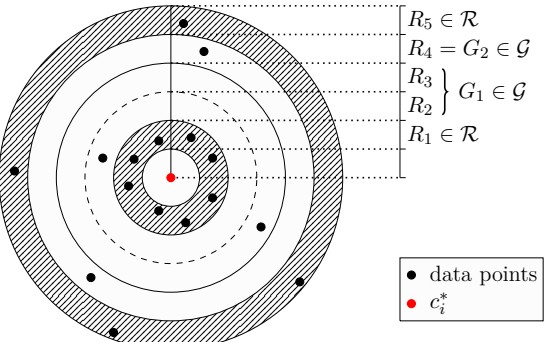

$R_5 \in \mathcal{R}$
$R_4 = G_2 \in \mathcal{G}$
$\left.\begin{array}{l} R_3 \\ R_2 \end{array}\right\} G_1 \in \mathcal{G}$
$R_1 \in \mathcal{R}$

• data points
• $c_i^*$

Figure 1: Illustration of Theorem 3.3 (plotted distance is the logarithm of the real distance).

### 3.1 THE FRAMEWORK OF BRAVERMAN ET AL. (2022)

Theorem 3.3 is a general geometric decomposition theorem for coresets which we use crucially. It partitions an arbitrary cluster into $\mathrm{poly}(k/\epsilon)$ *rings* and merge the remaining rings into $\mathrm{poly}(k/\epsilon)$ *groups* with low contribution to $\mathrm{cost}_z(X_i, c_i^*)$. (See Figure 1 for an illustration.)

**Theorem 3.3** (Decomposition into rings and groups (Braverman et al., 2022, Theorem 3.2)). *Let $X \subset \mathbb{R}^d$ be a set and $c \in \mathbb{R}^d$ be a center point. There exists an $O(nkd)$-time algorithm that computes a partition of $X$ into two disjoint collections of sets $\mathcal{R}$ and $\mathcal{G}$, such that $X = (\cup_{R \in \mathcal{R}} R) \cup (\cup_{G \in \mathcal{G}} G)$, where $\mathcal{R}$ is a collection of disjoint rings satisfying*

1. *$\forall R \in \mathcal{R}$, $R$ is a ring of the form $R = R_i(X, c)$ for some integer $i \in \mathbb{Z} \cup \{-\infty\}$, where $R_i(X, c) := X \cap \mathrm{ring}(c, 2^{i-1}, 2^i)$ for $i \in \mathbb{Z}$ and $R_{-\infty}(X, c) := X \cap \{c\}$*

2. *$|\mathcal{R}| \leq 2^{O(z \log z)} \cdot \tilde{O}(k\epsilon^{-z})$*

*and $\mathcal{G}$ is a collection of disjoint groups satisfying*

1. *$\forall G \in \mathcal{G}$, $G$ is the union of consecutive rings of $(X, c)$. Formally, $\forall G \in \mathcal{G}$, there exists two integers $-\infty \leq l_G \leq r_G$ such that $G = \cup_{i=l_G}^{r_G} R_i(X, c)$ and the intervals $\{[l_G, r_G], G \in \mathcal{G}\}$ are disjoint for different $G \in \mathcal{G}$*

2. *$|\mathcal{G}| \leq 2^{O(z \log z)} \cdot \tilde{O}(k\epsilon^{-z})$, and $\forall G \in \mathcal{G}$, $\mathrm{cost}_z(G, c) \leq (\frac{\epsilon}{6z})^z \cdot \frac{\mathrm{cost}_z(P, c)}{k \cdot \log(24z/\epsilon)}$.*

Rings and groups are inherently different geometric objects, hence they require different coreset construction methods.[2] As in Braverman et al. (2022), uniform sampling is applied on rings, but a *two-point coreset*, whose construction is defined in Definition 3.4, is applied for each group. Our main algorithm (Algorithm 1) also follows this general strategy.

**Definition 3.4** (Construction of two-point coreset (Braverman et al., 2022)). For a group $G \subset \mathbb{R}^d$ and a center point $c \in \mathbb{R}^d$, let $p_{\mathrm{far}}^G$ and $p_{\mathrm{close}}^G$ denote the furthest and closest point to $c$ in $G$. For every $p \in G$, compute the unique $\lambda_p \in [0, 1]$ such that $\mathrm{dist}^z(p, c) = \lambda_p \cdot \mathrm{dist}^z(p_{\mathrm{close}}^G, c) + (1 - \lambda_p) \cdot \mathrm{dist}^z(p_{\mathrm{far}}^G, c)$. Let $D_G = \{p_{\mathrm{far}}^G, p_{\mathrm{close}}^G\}$, $w_{D_G}(p_{\mathrm{close}}^G) = \sum_{p \in G} \lambda_p$, and $w_{D_G}(p_{\mathrm{far}}^G) = \sum_{p \in G}(1 - \lambda_p)$. $D_G$ is called the *two-point* coreset of $G$ with respect to $c$.

By definition, we can verify that $w_{D_G}(D_G) = |G|$ and $\mathrm{cost}_z(D_G, c) = \mathrm{cost}_z(G, c)$, which are useful for upper bounding the error induced by such two-point coresets.

---

[2]In Braverman et al. (2022), they mark some of the rings (which they call heavy rings), then group the remaining (unmarked) rings into groups. Our notion of ring corresponds to their "marked ring", and our group is the same as theirs. However, we do not need the concept of unmarked rings explicitly, since we only need to deal with the groups that are formed from them (and the construction of groups follows from a black box in Braverman et al. (2022)).

## 3.2 PROOF OF THEOREM 3.1

**Coreset Construction Algorithm**   We present our main algorithm in Algorithm 1. In Line 1 and Line 2, the set $L^*$ of outliers of $C^*$ is the set of $\gamma m$ furthest points to $C^*$ and $L^*$ is directly added into the coreset $S$. In Line 3 and Line 4, the inliers $P \setminus L^*$ are decomposed into $\beta k$ clusters with respect to $C^*$ and the linear time decomposition algorithm of Theorem 3.3 is applied in each cluster. In Line 5 and Line 6, similar to Braverman et al. (2022), a uniform sampling and a two-point coreset (see Definition 3.4) are applied in constructing coresets for rings and groups, respectively.

---

**Algorithm 1** Coreset Construction for $(k, z, m)$-ROBUST CLUSTERING

**Input:** dataset $P \subset \mathbb{R}^d$, $z \geq 1$, integer $k, m \geq 1$, an $(\alpha, \beta, \gamma)$-approximation $C^* = \{c_i^*\}_{i=1}^{\beta k}$
 1: let $L^* \leftarrow \arg\min_{|L|=\gamma m} \text{cost}_z(P \setminus L, C^*)$ denote the set of $\gamma m$ outliers
 2: add $L^*$ into $S$ and set $\forall x \in L^*, w_S \leftarrow 1$
 3: partition $P \setminus L^*$ into $\beta k$ clusters $P_1, ..., P_{\beta k}$ such that $P_i$ is the subset of $P \setminus L^*$ closest to $c_i^*$
 4: for each $i \in [\beta k]$, apply the decomposition of Theorem 3.3 to $(P_i, c_i^*)$ and obtain a collection $\mathcal{R}_i$ of disjoint rings and a collection $\mathcal{G}_i$ of disjoint groups
 5: for $i \in [\beta k]$ and every ring $R \in \mathcal{R}_i$, take a uniform sample $Q_R$ of size $2^{O(z \log z)} \cdot \tilde{O}(\frac{k}{\epsilon^{2z+2}})$ from $R$, set $\forall x \in Q_R, w_{Q_R}(x) \leftarrow \frac{|R|}{|Q_R|}$, and add $(Q_R, w_{Q_R})$ into $S$
 6: for $i \in [\beta k]$ and every group $G \in \mathcal{G}_i$ center $c_i^*$, construct a two-point coreset $(D_G, w_{D_G})$ of $G$ as in Definition 3.4 and add $(D_G, w_{D_G})$ into $S$
 7: return $(S, w_S)$

---

**Error Analysis**   Recall that $P$ is decomposed into 3 parts, the outliers $L^*$, the collection of rings, and the collection of groups. We prove the coreset property for each of the 3 parts and claim the union yields an $\epsilon$-coreset of $P$ for $(k, z, m)$-ROBUST CLUSTERING. As $L^*$ is identical in the data set $P$ and the coreset $S$, we only have to put effort in the rings and groups. We first introduce the following relaxed coreset definition which allows additive error.

**Definition 3.5.** Let $P \subset \mathbb{R}^d$, $0 < \epsilon < 1$ and $A \geq 0$, a weighted set $(S, w_S)$ is an $(\epsilon, A)$-coreset of $X$ for $(k, z, t)$-ROBUST CLUSTERING if for every $C \subset \mathbb{R}^d, |C| = k$,

$$|\text{cost}_z^{(t)}(P, C) - \text{cost}_z^{(t)}(S, C)| \leq \epsilon \cdot \text{cost}_z^{(t)}(P, C) + \epsilon \cdot A.$$

This allowance of additive error turns out to be crucial in our analysis, and eventually we are able to charge the total additive error to the (near-)optimal cost, which enables us to obtain the coreset (without additive error).

The following two are the key lemmas for the proof of Theorem 3.1, where we analyze the guarantee of the uniform-sampling coresets for rings (Lemma 3.6) and the two-point coresets (Lemma 3.7).

**Lemma 3.6** (Coresets for rings). *Let $Q = \bigcup_{i \in [\beta k]} \bigcup_{R \in \mathcal{R}_i} Q_R$ denote the coreset of the rings $R_{\text{all}} = \bigcup_{i \in [\beta k]} \bigcup_{R \in \mathcal{R}_i} R$, constructed by uniform sampling as in Line 5 of Algorithm 1, then $\forall t, 0 \leq t \leq m$, $Q$ is an $(\epsilon, \text{cost}_z(R_{\text{all}}, C^*))$-coreset of $R_{\text{all}}$ for $(k, z, t)$-ROBUST CLUSTERING.*

*Proof.* The proof can be found in Section F. □

**Lemma 3.7** (Two-point coresets for groups). *Let $D = \bigcup_{i \in [\beta k]} \bigcup_{G \in \mathcal{G}_i} D_G$ denote the two-point coresets of the groups $G_{\text{all}} = \bigcup_{i \in [\beta k]} \bigcup_{G \in \mathcal{G}_i} G$, as in Line 6 of Algorithm 1, then for every $t, 0 \leq t \leq m$, $D$ is an $(\epsilon, \text{cost}_z(P \setminus L^*, C^*))$-coreset of $G_{\text{all}}$ for $(k, z, t)$-ROBUST CLUSTERING.*

*Proof.* The proof can be found in Section G. □

*Proof of Theorem 3.1.* Fix a center $C \subset \mathbb{R}^d, |C| = k$ and fix a $t \in [0, m]$, we first prove that

$$\text{cost}_z^{(t)}(S, C) \leq (1 + \epsilon) \text{cost}_z^{(t)}(P, C) + \epsilon \cdot \text{cost}_z(P \setminus L^*, C^*).$$

To this end, assume $L \subset P$ is the set of outliers for $C$ with $|L| = t$.

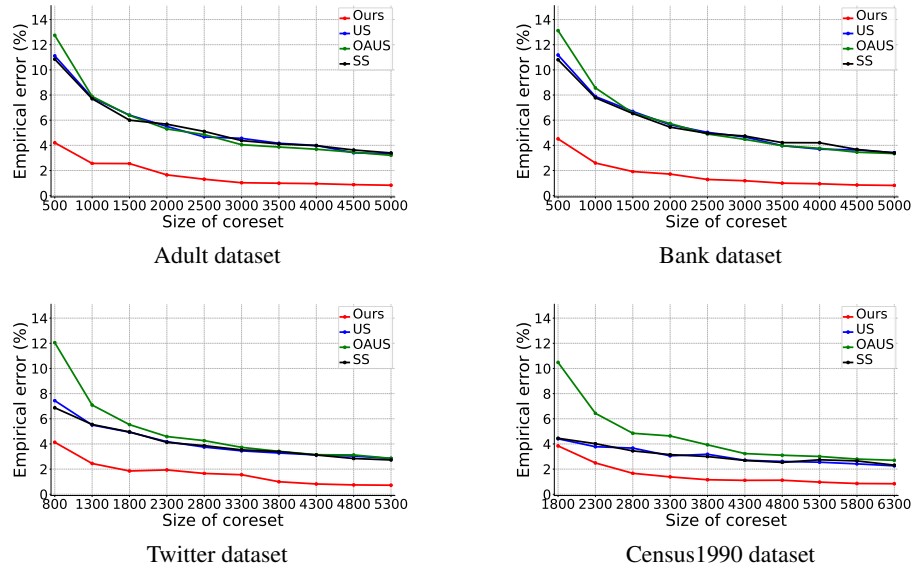

Figure 2: The tradeoff between the coreset size and the empirical error.

Let $t_R = |L \cap R_{\text{all}}|, t_G = |L \cap G_{\text{all}}|$. By Lemma 3.6 and Lemma 3.7, there exists weighted subset $T_Q \subset Q, T_D \subset D$ such that, $w_{T_Q}(T_Q) = t_R, w_{T_D}(T_D) = t_G$,

$$\text{cost}_z(Q - T_Q, C) \le (1 + \epsilon) \text{cost}_z(R_{\text{all}} - (L \cap R_{\text{all}}), C) + \epsilon \cdot \text{cost}_z(R_{\text{all}}, C^*) \tag{3}$$

and

$$\text{cost}_z(D - T_D, C) \le (1 + \epsilon) \text{cost}_z(G_{\text{all}} - (L \cap G_{\text{all}}), C) + \epsilon \cdot \text{cost}_z(P \setminus L^*, C^*) \tag{4}$$

Define a weighted subset $(T, w_T)$ of $S$, such that $T = (L \cap L^*) \cup T_Q \cup T_D$. Then $w_T(T) = t$ and

$$
\begin{aligned}
\text{cost}_z^{(t)}(S, C) &\le & \text{cost}_z(S - T, C) \\
&= & \text{cost}_z(L^* - (L \cap L^*), C) + \text{cost}_z(Q - T_Q, C) + \text{cost}_z(D - T_D, C) \\
&\le & \text{cost}_z(L^* - (L \cap L^*), C) + (1 + \epsilon) \text{cost}_z(R_{\text{all}} - (L \cap R_{\text{all}}), C) \\
&+ & \epsilon \cdot \text{cost}_z(R_{\text{all}}, C^*) + (1 + \epsilon) \text{cost}_z(G_{\text{all}} - (L \cap G_{\text{all}}), C) \\
&+ & \epsilon \cdot \text{cost}_z(P \setminus L^*, C^*) \\
&\le & (1 + \epsilon) \text{cost}_z(P - L, C) + O(\epsilon) \cdot \text{cost}_z(P \setminus L^*, C^*) \\
&\le & (1 + O(\alpha \cdot \epsilon)) \text{cost}_z^{(t)}(P, C).
\end{aligned}
$$

Similarly, we can also obtain that $\text{cost}_z^{(t)}(P, C) \le (1 + O(\alpha \cdot \epsilon)) \text{cost}_z^{(t)}(S, C)$ for any $0 \le t \le m$. It remains to scale $\epsilon$ by a universal constant.

We analyze the time complexity. Clearly, the running time of Algorithm 1 is dominated by the first four lines, each of which takes $O(nkd)$ time. Apart from the steps of building the coresets, the time for the initial tri-criteria approximation is discussed in Section 1.3 and Section A. □

## 4 EXPERIMENTS

We implement our coreset construction algorithm and evaluate its empirical performance on various real datasets. We compare it with several baselines and demonstrate the superior performance of our coreset. In addition, we show that our coresets can significantly speed up approximation algorithms for both $(k, m)$-ROBUST MEDIAN and $(k, m)$-ROBUST MEANS problems.

**Experiment Setup** Our experiments are conducted on publicly available clustering datasets, see Table 1 for a summary of specifications and choice of parameters. For all datasets, we select numerical features to form a vector in $\mathbb{R}^d$ for each record. For larger dataset, particularly Census1990

Table 1: Specifications of datasets and the choice of the parameters.

| dataset | size | subsample | dim. | # of outliers $m$ |
|---|---|---|---|---|
| Adult  (Dua & Graff, 2017) | 48842 | - | 6 | 200 |
| Bank  (Moro et al., 2014) | 41188 | - | 10 | 200 |
| Twitter (Chan et al., 2018) | 21040936 | $10^5$ | 2 | 500 |
| Census1990 (Meek et al., 1990) | 2458285 | $10^5$ | 68 | 1500 |

and Twitter, we subsample it to $10^5$ points so that inefficient baselines can still finish in a reasonable amount of time. Unless otherwise specified, we typically set $k = 5$ for the number of centers. The number of outliers $m$ is determined by a per-dataset basis, via observing the distance distribution of points to a near-optimal center (see Section C for details). All experiments are conducted on a PC with Intel Core i7 CPU and 16 GB memory, and algorithms are implemented using C++ 11. We implement our coreset following Algorithm 1 except for a few modifications. The detailed modifications are described in Section C.

**Empirical Error**   We evaluates the tradeoff between the coreset size and empirical error under the $(k, m)$-ROBUST MEDIAN objective. In general, for $(k, z, m)$-ROBUST CLUSTERING, given a coreset $S$, define its empirical error, denoted as $\hat{\epsilon}(S, C)$, for a specific center $C \subset \mathbb{R}^d$, $|C| = k$ as $\hat{\epsilon}(S, C) := \frac{|\text{cost}_z^{(m)}(X,C) - \text{cost}_z^{(m)}(S,C)|}{\text{cost}_z^{(m)}(X,C)}$. Since it is difficult to exactly verify whether a coreset preserves the objective for *all* centers (as required by the definition), we evaluate the empirical error, denoted as $\hat{\epsilon}(S)$, for the coreset $S$ as the maximum empirical error over $\mathcal{C}$, which is a collection of 500 randomly-chosen center sets, i.e., $\hat{\epsilon}(S) := \max_{C \in \mathcal{C}} \hat{\epsilon}(S, C)$. Note that $\hat{\epsilon}(S)$ is defined in a way similar to the worst-case error parameter $\epsilon$ as in Definition 2.1.

**Baselines**   We compare our coreset with the following baselines: a) uniform sampling (US), where we draw $N$ independent uniform samples from $X$ and set the weight $\frac{|X|}{N}$ for each sample, b) outlier-aware uniform sampling (OAUS), where we follow Line 1 - Line 2 of Algorithm 1 to add $m$ outliers $L^*$ to the coreset and sample $N - m$ data points from $X \setminus L^*$ as in US baseline, and c) sensitivity sampling (SS), the previous coreset construction algorithm of Feldman & Schulman (2012).

**Experiment: Size-error Tradeoff**   For each coreset algorithm, we run it to construct coresets of varying target sizes $N$, ranging from $m + 300$ to $m + 4800$, with a step size of 500. We evaluate the empirical error $\hat{\epsilon}(\cdot)$ and we plot the size-error curves in Figure 2 for each baseline and dataset. To make the measurement stable, the coreset construction and evaluations are run 100 times independently and the average is reported. As can be seen from Figure 2, our coreset admits a similar error curve regardless of the dataset, and it achieves about 2.5% error using a coreset of size $m + 800$ (within 2.3% - 2.5% of data size), which is perfectly justified by our theory that the coreset size only depends on $O(m + \text{poly}(k\epsilon^{-1}))$. Our coresets outperform all three baselines by a significant margin in every dataset and every target coreset size. Interestingly, the two baselines SS and US seem to perform similarly, even though the construction of SS (Feldman & Schulman, 2012) is way more costly since its running time has an exponential dependence on $k + m$, which is already impractical in our setting of parameters. Another interesting finding is that, OAUS performs no better than US overall, and both are much worse than ours. This indicates that it is not the added initial outliers $L^*$ (as in Algorithm 1) that leads to the superior performance of our coreset. Finally, we also observe that our coreset has a smaller variance in the empirical error ($\approx 10^{-6}$), compared with other baselines ($\approx 10^{-4}$).

**Experiment: Impact of The Number of Outliers**   We also examine the impact of the number of outliers $m$ on the empirical error. The details of this experiment can be found in Section D.

**Experiment: Speeding Up Existing Approximation Algorithms**   We validate the ability of our coresets for speeding up existing approximation algorithms for robust clustering. Due to space limit, the details and results can be found in Section E.

ACKNOWLEDGMENTS

Research is partially supported by a national key R&D program of China No. 2021YFA1000900, a startup fund from Peking University, and the Advanced Institute of Information Technology, Peking University.

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

# Appendices

## A  ALGORITHMS FOR TRI-CRITERIA APPROXIMATION

Various known algorithms that offer different tradeoffs may be used for the required $(\alpha, \beta, \gamma)$-approximation. In particular, Friggstad et al. (2019) designed a polynomial-time $\mathcal{A}(n, k, d, z) = n^{O(1)}$ algorithm with $\alpha = O(2^z), \beta = O(1), \gamma = 1$; Bhaskara et al. (2019) gave a near-linear time $\mathcal{A}(n, k, d, z) = \tilde{O}(nkd)$ algorithm with $\alpha = O(2^{O(z)}), \beta = O(1), \gamma = O(1)$ (which implies the statement in Theorem 1.1).[3] Finally, true approximation algorithms, i.e., $\beta = \gamma = 1$, are known for both $(k, m)$-ROBUST MEDIAN and $(k, m)$-ROBUST MEANS, and they run in polynomial-time $\mathcal{A}(n, k, d, z) = n^{O(1)}$ and achieves $\alpha = O(1)$ (Chen, 2008; Krishnaswamy et al., 2018).

## B  TECHNICAL LEMMAS

**Lemma B.1** (Generalized triangle inequalities). *Let $a, b \geq 0$ and $\delta \in (0, 1)$, then for $z \geq 1$,*

1. *(Lemma A.1 of Makarychev et al. (2019))* $(a + b)^z \leq (1 + \delta)^{z-1} \cdot a^z + (1 + \frac{1}{\delta})^{z-1} \cdot b^z$

2. *(Claim 5 of Sohler & Woodruff (2018))* $(a + b)^z \leq (1 + \delta) \cdot a^z + (\frac{3z}{\delta})^{z-1} \cdot b^z$

The following lemma is a simple but useful way to bound the error between coresets and data sets and the proof idea is similar to Lemma 3.5 of Braverman et al. (2022) which relies on coupling the mass and applying generalized triangle inequality Lemma B.1. This technical lemma is used in many places in our entire proof.

**Lemma B.2.** *Let $B \subset P_i$ be either a ring or a group. Assume $(U, w_U)$ and $(V, w_V)$ are two weighted subsets of $B$ such that $w_U(U) = w_V(V) = N$, then for every $C \subset \mathbb{R}^d, |C| = k$ we have*

$$|\cost_z(U, C) - \cost_z(V, C)| \leq \epsilon \cdot \cost_z(U, C) + (\frac{6z}{\epsilon})^{z-1} \cdot \big(\cost_z(U, c_i^*) + \cost_z(V, c_i^*)\big). \quad (5)$$

*Proof.* Since $w_U(U) = w_V(V)$, there must exist a matching $M : U \times V \to \mathbb{R}_{\geq 0}$ between the mass of $U$ and $V$. So $\forall u \in U, \sum_{v \in V} M(u, v) = w_U(u)$ and $\forall v \in V, \sum_{u \in U} M(u, v) = w_V(v)$. By generalized triangle inequality Lemma B.1 we have,

$$
\begin{aligned}
&|\cost_z(U, C) - \cost_z(V, C)| \\
\leq\ & \sum_{u \in U} \sum_{v \in V} M(u, v)|\dist(u, C)^z - \dist(v, C)^z| \\
\leq\ & \sum_{u \in U} \sum_{v \in V} M(u, v)\big(\epsilon \cdot \dist(u, C)^z + (\frac{3z}{\epsilon})^{z-1} \cdot (\dist(u, C) - \dist(v, C))^z\big) \\
\leq\ & \epsilon \sum_{u \in U} w_U(u) \cdot \dist(u, C)^z + (\frac{3z}{\epsilon})^{z-1} \cdot \sum_{u \in U} \sum_{v \in V} M(u, v) \cdot (\dist(u, c_i^*) + \dist(v, c_i^*))^z \\
\leq\ & \epsilon \cdot \cost_z(U, C) + (\frac{3z}{\epsilon})^{z-1} \cdot \big(\sum_{u \in U} w_U(u) \cdot 2^{z-1} \cdot \dist(u, c_i^*)^z + \sum_{v \in V} w_V(v) \cdot 2^{z-1} \cdot \dist(v, c_i^*)\big) \\
\leq\ & \epsilon \cdot \cost_z(U, C) + (\frac{6z}{\epsilon})^{z-1}\big(\cost_z(U, c_i^*) + \cost_z(V, c_i^*)\big)
\end{aligned}
$$

□

## C  MORE DETAILS OF THE EXPERIMENT

**Determining The Number of Outliers**  To determine the number of outliers $m$ for each (subsampled) dataset in our experiment, we run a vanilla $k$-MEANS clustering (without outliers) algorithm,

---

[3]Bhaskara et al. (2019) only showed the case of $z = 2$, but we check that it also generalizes to other $z$'s.

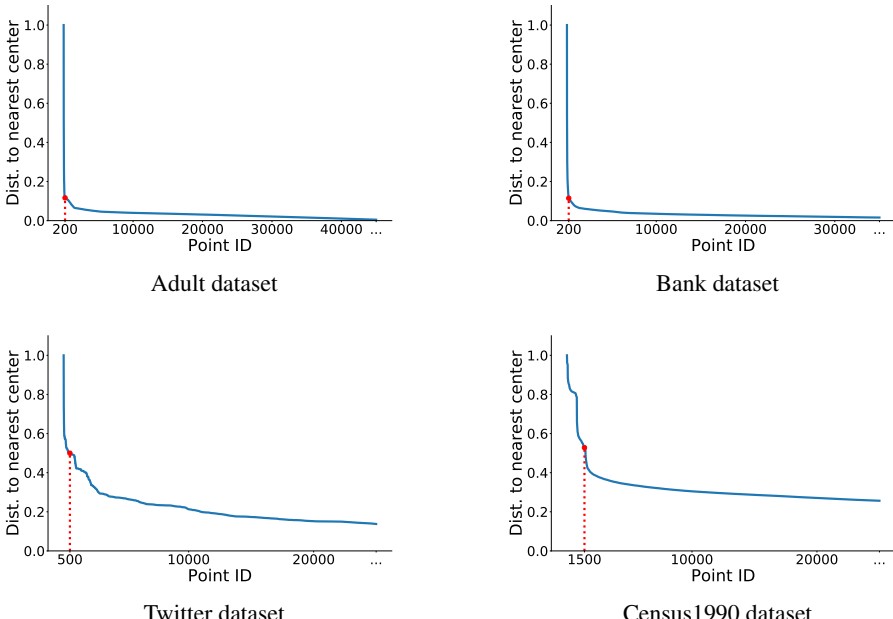

Figure 3: Distances to the found near-optimal center (using a vanilla clustering algorithm) for each point, sorted decreasingly and rescaled to $[0, 1]$.

and plot the distribution of distances from data points to the found near-optimal centers. As shown in Figure 3, every dataset admits a clear breaking point that defines the outliers, and we pick $m$ accordingly.

**Implementation Details** Our coreset implementation mostly follows Algorithm 1 except for a few modifications. For efficiency, we use a near-linear time algorithm by Bhaskara et al. (2019) to compute an $(O(1), O(1), O(1))$-approximation (as required by Algorithm 1), but we still add $m$ outliers to coreset (in Line 2) instead of adding all the found ones. Moreover, since it is more practical to directly set the target coreset size $N$ (instead of solving for $N$ from $\epsilon$), we modify the algorithm so that the generated coreset has exactly $N$ points. Specifically, the coreset size is affected by two key parameters, one is a threshold, denoted as $t$, used to determine how the rings and groups are formed in the construction of Theorem 3.3 (whose details can be found in Braverman et al. (2022)), and the other, denoted as $s$, is the size of each uniform sample (used in Line 5). Here, we heuristically set $t = O(\frac{1}{N-m})$ and solve for $s$ such that the total size equals to $N$.

## D  EXPERIMENT: IMPACT OF THE NUMBER OF OUTLIERS

We examine the impact of the number of outliers $m$ on empirical error. Specifically, we experiment with varying $m$, but a fixed $N - m$, which is the number of "samples" besides the included outliers $L^*$ in our algorithm. We pick a typical value of $N - m = 800$ based on the curves of Figure 2, . We plot this outlier-error curve in Figure 4, and we observe that while some of our baselines have a fluctuating empirical error, the error curve of our coreset is relatively stable. This suggests that the empirical error of our coreset is mainly determined by the number of additional samples $N - m$, and is mostly independent of the number of outliers $m$ itself.

## E  EXPERIMENT: SPEEDING UP EXISTING APPROXIMATION ALGORITHMS

We validate the ability of our coresets for speeding up existing approximation algorithms for robust clustering. We consider two natural algorithms and run them on top of our coreset for speedup: a Lloyd-style algorithm tailored to $(k, m)$-ROBUST MEANS (Chawla & Gionis, 2013) seeded by

Table 2: Running time and costs for LL and LS with/without coresets. $T_X$ and $T_S$ are the running time without/with the coreset, respectively. Similarly, cost and cost$'$ are the clustering costs without/with the coreset. $T_C$ is coreset construction time. This entire experiment is repeated 10 times and the average is reported.

| dataset | algorithm | cost | cost$'$ | $T_C$ (s) | $T_S$ (s) | $T_X$ (s) |
|---------|-----------|------|---------|-----------|-----------|-----------|
| Adult | LL | $3.790 \times 10^{13}$ | $3.922 \times 10^{13}$ | 0.4657 | 0.06385 | 16.51 |
| | LS | $1.100 \times 10^{9}$ | $1.107 \times 10^{9}$ | 0.5300 | 1.147 | 204.8 |
| Bank | LL | $4.444 \times 10^{8}$ | $4.652 \times 10^{8}$ | 0.4399 | 0.05900 | 11.40 |
| | LS | $4.717 \times 10^{6}$ | $4.721 \times 10^{6}$ | 0.4953 | 1.220 | 186.6 |
| Twitter | LL | $3.218 \times 10^{7}$ | $3.236 \times 10^{7}$ | 0.9493 | 0.08289 | 11.27 |
| | LS | $1.476 \times 10^{6}$ | $1.451 \times 10^{6}$ | 1.064 | 2.135 | 460.2 |
| Census1990 | LL | $1.189 \times 10^{7}$ | $1.208 \times 10^{7}$ | 3.673 | 0.4809 | 40.54 |
| | LS | $1.165 \times 10^{6}$ | $1.163 \times 10^{6}$ | 4.079 | 24.83 | 2405 |

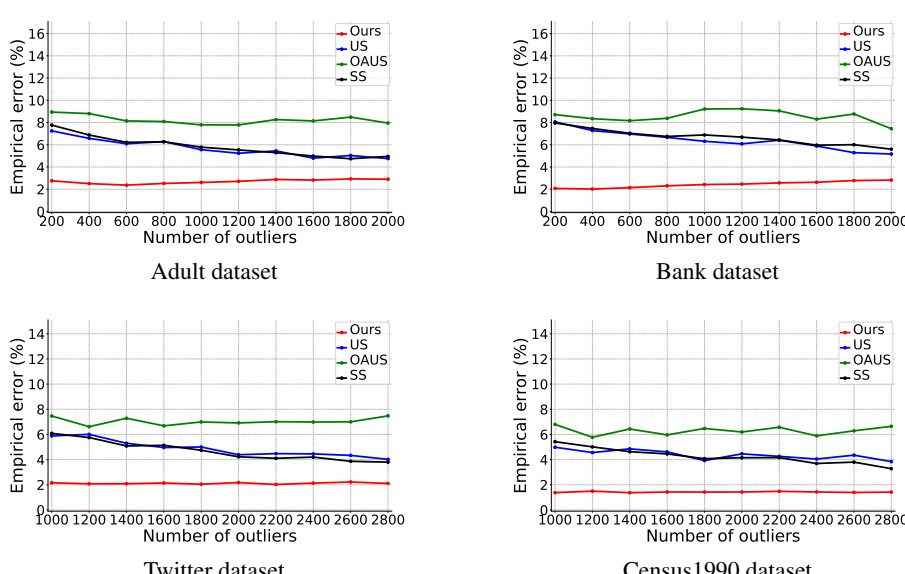

Figure 4: The impact of the number of outliers $m$ on the empirical error

a modified $k$-MEANS++ for robust clustering (Bhaskara et al., 2019), which we call "LL", and a local search algorithm for $(k, m)$-ROBUST MEDIAN (Friggstad et al., 2019), which we call "LS". We note that for LS, we uniformly sample 100 points from the dataset and use them as the only potential centers, since otherwise it takes too long to run on the original dataset (without coresets). We use a coreset of size $m + 500$ for each dataset (recalling that $m$ is picked per dataset according to Figure 3) to speed up the algorithms. To make a consistent comparison, we measure the clustering costs on the original dataset for all runs (instead of on the coreset).

We report in Table 2 the running time and the cost achieved by LL and LS, with and without coresets. The results show that the error incurred by using coreset is tiny ($< 5\%$ error), but the speedup is a significant 80x-250x for LL, and a 100x-200x for LS. Even taking the coreset construction time into consideration, it still achieves a 10x-30x speedup to LL and a 80x-140x speedup to LS. We conclude that our coreset drastically improves the running time for existing approximation algorithms, while only suffering a neglectable error.

# F  PROOF OF LEMMA 3.6: ERROR ANALYSIS OF UNIFORM SAMPLING

As with recent works in Euclidean coresets (Cohen-Addad et al., 2021a;b; 2022; Braverman et al., 2022), we make use of an iterative size reduction Braverman et al. (2021) and a terminal embedding

technique Narayanan & Nelson (2019), which allows us to trade the factor $O(d)$ in coreset size bound with a factor of $O\left(\frac{\log(k/\epsilon)}{\epsilon^2}\right)$. Hence, it suffices to prove that a uniform sample of size $\tilde{O}(\frac{kd}{\epsilon^{2z}})$ yields the desired coreset.

The following simple formula can be obtained via integration by parts.

**Fact F.1.** *Let $(Y, w_Y)$ denote a weighted dataset and $C \subseteq \mathbb{R}^d, |C| = k$ then for every $0 \le t \le w_Y(Y)$*

$$\text{cost}_z^{(t)}(P, C) = \int_0^\infty z \cdot u^{z-1} \cdot \left( w_Y(Y) - m - w_Y\big(\text{Balls}(C, u) \cap Y\big) \right)^+ du.$$

The following notion of $\epsilon$-approximation for $k$-balls range space is well-studied in PAC learning and computational geometry communities (see e.g. Har-peled (2011)).

**Definition F.2** ($\epsilon$-Approximation for $k$-balls range space)**.** Let $\text{Balls}_k = \{\text{Balls}(C, u) \mid C \subset \mathbb{R}^d, |C| = k, u > 0\}$ denote the set of unions of $k$ balls with the same radius. For a dataset $P \subset \mathbb{R}^d$, the $k$-Balls range space on $P$ is denoted by $(P, \mathcal{P}_k)$ where $\mathcal{P}_k := \{P \cap \text{Balls}_k(C, u) \mid \text{Balls}_k(C, u) \in \text{Balls}_k\}$. A subset $Y \subset P$ is called an $\epsilon$-approximation of the $k$-Balls range space $(P, \mathcal{P}_k)$ if for every $\text{Balls}(C, u) \in \text{Balls}_k$,

$$\left| \frac{|P \cap \text{Balls}(C, u)|}{|P|} - \frac{|Y \cap \text{Balls}(C, u)|}{|Y|} \right| \le \epsilon.$$

The following lemma reduces the construction of an $\epsilon$-approximation to uniform sampling.

**Lemma F.3** (Li et al. (2001))**.** *Assume $Q$ is a uniform sample of size $\tilde{O}(\frac{kd}{\epsilon^2})$ from $P$, then with probability at least $1 - \frac{1}{\text{poly}(k/\epsilon)}$, $Q$ is an $\epsilon$-approximation of the $k$-Balls range space on $P$.*

The following Lemma F.4 shows an $(\frac{\epsilon}{12z})^z$-approximation yields a $2^{O(z \log z)} \cdot \epsilon$-coreset for robust $(k, z, t)$-ROBUST CLUSTERING for every $t$.

**Lemma F.4.** *Assume $R = P_i \cap \text{ring}(c_i^*, r, 2r)$ is a ring in the cluster $P_i$. Let $Q_R$ be an $(\frac{\epsilon}{12z})^z$-approximation of the $k$-balls range space on $R$. Suppose every element of $Q_R$ is re-weighted by $\frac{|R|}{|Q_R|}$ then for every $C \subset \mathbb{R}^d, |C| = k$ and every $t, 0 \le t \le \min\{|R|, m\}$,*

$$| \text{cost}_z^{(t)}(R, C) - \text{cost}_z^{(t)}(Q_R, C)| \le \epsilon \cdot \text{cost}_z^{(t)}(R, C) + \epsilon r^z |R|. \tag{6}$$

*Proof.* Fix a $C \subset \mathbb{R}^d, |C| = k$. As $Q_R$ is an $(\frac{\epsilon}{12z})^z$-approximation of the $k$-Balls range space on $R$, we know that for every $u > 0$,

$$|w_R\big(\text{Balls}(C, u) \cap R\big) - w_{Q_R}\big(\text{Balls}(C, u) \cap Q_R\big)| \le (\frac{\epsilon}{12z})^z \cdot |R|. \tag{7}$$

Let $T_{\text{close}} = \min_{x \in R} \text{dist}(x, C)$ and $T_{\text{far}} = \max_{x \in R} \text{dist}(x, C)$. Since $R \subset \text{ring}(c_i^*, r, 2r)$, the diameter of $R$ is at most $4r$ and this implies $T_{\text{far}} - T_{\text{close}} \le 4r$. Since $Q_R$ is a subset of $R$, we know that for every $u \notin [T_{\text{close}}, T_{\text{far}}]$,

$$w_R\big(\text{Balls}(C, u) \cap R\big) = w_{Q_R}\big(\text{Balls}(C, u) \cap Q_R\big) \tag{8}$$

To prove (6), we do the following case analysis.

If the number of outliers $t \ge (1 - (\frac{\epsilon}{12z})^z) \cdot |R|$, let $L_R \subset R$ and $L_Q \subset Q_R$ denote the outliers of $R$ and $Q$ with respect to $C$. Using Lemma B.2, we know that

$$
\begin{aligned}
& | \text{cost}_z^{(t)}(R, C) - \text{cost}_z^{(t)}(Q_R, C)| \\
\le \quad & \epsilon \cdot \text{cost}_z^{(t)}(R, C) + (\frac{6z}{\epsilon})^{z-1} \cdot \big( \text{cost}_z(R - L_R, c_i^*) + \text{cost}_z(Q_R - L_Q, c_i^*)\big) \\
\le \quad & \epsilon \cdot \text{cost}_z^{(t)}(R, C) + (\frac{6z}{\epsilon})^{z-1} \cdot (\frac{\epsilon}{12z})^z \cdot |R| \cdot (2r)^z \\
\le \quad & \epsilon \cdot \text{cost}_z^{(t)}(R, C) + \epsilon r^z |R|.
\end{aligned}
$$

If $t < (1 - (\frac{\epsilon}{12z})^z) \cdot |R|$, using (7), (8), Fact F.1 and the generalized triangle inequality Lemma B.1, we have,

$$
\begin{aligned}
& |\operatorname{cost}_z^{(t)}(P, C) - \operatorname{cost}_z^{(t)}(Q_R, C)| \\
\leq\ & \int_{T_{\mathrm{close}}}^{T_{\mathrm{far}}} z u^{z-1} \cdot \big| w_R\big(\operatorname{Balls}(C, u) \cap R)\big) - w_{Q_R}\big(\operatorname{Balls}(C, u) \cap Q_R\big)\big| du \\
\leq\ & (\frac{\epsilon}{12z})^z \cdot |R| \cdot (T_{\mathrm{far}}^z - T_{\mathrm{close}}^z) \\
\leq\ & (\frac{\epsilon}{12z})^z \cdot |R| \cdot \big(\epsilon \cdot T_{\mathrm{close}}^z + (\frac{3z}{\epsilon})^{z-1} \cdot (4r)^z\big) \\
\leq\ & \epsilon \cdot (\frac{\epsilon}{12z})^z \cdot |R| \cdot T_{\mathrm{close}}^z + \epsilon r^z |R| \\
\leq\ & \epsilon \cdot \operatorname{cost}_z^{(t)}(R, C) + \epsilon r^z |R|
\end{aligned}
$$

where for the last inequality, we have used the fact that

$$
\operatorname{cost}_z^{(t)}(R, C) \geq (|R| - t) \cdot T_{\mathrm{close}}^z \geq (\frac{\epsilon}{12z})^z \cdot |R| \cdot T_{\mathrm{close}}^z.
$$

□

We are ready to prove Lemma 3.6.

*Proof of Lemma 3.6.* Fix a center $C \subset \mathbb{R}^d, |C| = k$. By Lemma F.3, the sample size in Line 5 of Algorithm 1 implies that $Q_R$ is an $(\frac{\epsilon}{12z})^z$-approximation of the $k$-balls range space on $R$ for every $R \in \bigcup_{i \in [\beta k]} \mathcal{R}_i$. By lemma F.4 and the union bound, with probability at least 0.9, for every $i \in [\beta]$, for every ring $R \in \mathcal{R}_i$, and for every $e \in [0, |R|]$,

$$
|\operatorname{cost}_z^{(e)}(R, C) - \operatorname{cost}_z^{(e)}(Q_R, C)| \leq \epsilon \cdot \operatorname{cost}_z^{(e)}(R, C) + \epsilon \cdot \operatorname{cost}_z(R, c_i^*). \tag{9}
$$

Let $L$ denote the set of $t$ outliers of $R_{\mathrm{all}}$ with respect to $C$. By (9), for every $R \in \bigcup_{i \in [\beta k]} \mathcal{R}_i$, there exists a weighted subset $T_R \subset Q_R$ such that $w_{T_R}(T_R) = w_L(L \cap R)$ and

$$
\operatorname{cost}_z(Q_R - T_R, C) \leq (1 + \epsilon) \cdot \operatorname{cost}_z(R - (L \cap R), C) + \epsilon \cdot \operatorname{cost}_z(R, c_i^*).
$$

Summing over all $R \in \bigcup_{i \in [\beta k]} \mathcal{R}_i$, we know that,

$$
\begin{aligned}
& \operatorname{cost}_z^{(t)}(Q, C) \\
\leq\ & \sum_{i \in [\beta k]} \sum_{R \in \mathcal{R}_i} \operatorname{cost}_z(Q_R - T_R, C) \\
\leq\ & \sum_{i \in [\beta k]} \sum_{R \in \mathcal{R}_i} \big((1 + \epsilon) \cdot \operatorname{cost}_z(R - (L \cap R), C) + \epsilon \cdot \operatorname{cost}_z(R, c_i^*)\big) \\
=\ & (1 + \epsilon) \cdot \operatorname{cost}_z(R_{\mathrm{all}} - L, C) + \epsilon \cdot \operatorname{cost}_z(R_{\mathrm{all}}, C^*) \\
=\ & (1 + \epsilon) \cdot \operatorname{cost}_z^{(t)}(R_{\mathrm{all}}, C) + \epsilon \cdot \operatorname{cost}_z(R_{\mathrm{all}}, C^*).
\end{aligned}
$$

On the same way, we can show that

$$
\operatorname{cost}_z^{(t)}(R_{\mathrm{all}}, C) \leq (1 + \epsilon) \cdot \operatorname{cost}_z^{(t)}(Q, C) + \epsilon \cdot \operatorname{cost}_z(R_{\mathrm{all}}, C^*).
$$

Thus we finish the proof. □

## G  PROOF OF LEMMA 3.7: ERROR ANALYSIS OF TWO-POINT CORESETS

Throughout this section, we fix a center set $C \subset \mathbb{R}^d, |C| = k$ and prove the coreset property of $D$ with respect to $C$. To analyze the error of two-point coreset for $G_{\mathrm{all}}$, we further decomposes all groups into colored groups and uncolored groups based on the position of $C$ in the following Lemma G.1, which was also considered in Braverman et al. (2022). Furthermore, inside our proof, we also consider a more refined type of groups called *special groups*. An overview illustration of these groups and other relevant notions can be found in Figure 5.

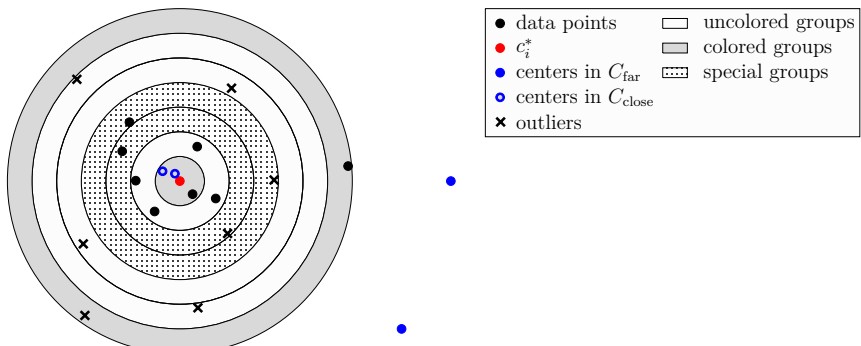

Figure 5: An illustration of the decomposition into colored, uncolored and special groups with respect to $C = C_{\text{far}} \cup C_{\text{close}}$, where the radii of balls are taken the logarithm.

**Lemma G.1** (Colored groups and uncolored groups (Braverman et al., 2022))**.** *For a center set $C \subset \mathbb{R}^d, |C| = k$, a collection of groups $\mathcal{G}_i$ can be further divided into* colored *groups and* uncolored *groups with respect to $C$ such that*

1. *there are at most $O(k \log \frac{z}{\epsilon})$ colored groups and*

2. *for every uncolored group $G \in \mathcal{G}_i$, for every $u \in C$, either $\forall p \in G, \text{dist}(u, c_i^*) < \frac{\epsilon}{9z} \cdot \text{dist}(p, c_i^*)$ or $\forall p \in G, \text{dist}(u, c_i^*) > \frac{24z}{\epsilon} \text{dist}(p, c_i^*)$.*

Let $G \in \mathcal{G}_i$ be an uncolored group with respect to $C$, Lemma G.1 implies that the center set $C$ can be decomposed into a "close" portion and a "far" portion to $G$, as in the following Definition G.2.

**Definition G.2** (Braverman et al. (2022))**.** For a center set $C$, assume $G \in \mathcal{G}_i$ is an uncolored group with respect to $C$. Define

$$C_{\text{far}}^G = \{u \in C \mid \forall p \in G, \text{dist}(u, c_i^*) > \frac{24z}{\epsilon} \text{dist}(p, c_i^*)\},$$

and

$$C_{\text{close}}^G = \{u \in C \mid \forall p \in G, \text{dist}(u, c_i^*) < \frac{\epsilon}{9z} \cdot \text{dist}(p, c_i^*)\}.$$

Remark that $C = C_{\text{far}}^G \cup C_{\text{close}}^G$ by the property of uncolored group.

The following Lemma G.3 shows the difference of cost to any center $C \subset \mathbb{R}^d, |C| = k$ between a group $G$ and its two-point coreset $D_G$ can always be bounded by a small additive error, via generalized triangle inequality Lemma B.1.

By combining Lemma B.2 and the fact that $\text{cost}_z(G_i, c_i^*) \le (\frac{\epsilon}{6z})^z \cdot \frac{\text{cost}_z(P_i, c_i^*)}{k \log(24z/\epsilon)}$, we can obtain the following inequality.

**Lemma G.3** (Robust variant of (Braverman et al., 2022, Lemma 3.5))**.** *For a group $G \in \mathcal{G}_i$, assume $(U, w_U)$ and $(V, w_V)$ are two weighted subsets of $G$ such that $w_U(U) = w_V(V)$. Then for every $C \subset \mathbb{R}^d, |C| = k$,*

$$|\text{cost}_z(U, C) - \text{cost}_z(V, C)| \le \epsilon \cdot \text{cost}_z(U, C) + \epsilon \cdot \frac{\text{cost}_z(P_i, c^*)}{2k \log(z/\epsilon)}. \tag{10}$$

**Lemma G.4.** *Let $G$ denote an uncolored group with respect to $C$. Suppose $(U, w_U)$ and $(V, w_V)$ are two weighted subsets of $G$ such that one of the following items hold,*

1. *either $C_{\text{close}}^G \ne \emptyset$ and $\text{cost}_z(U, c_i^*) = \text{cost}_z(V, c_i^*)$,*

2. *or $C_{\text{close}}^G = \emptyset$ and $w_U(U) = w_V(V)$.*

*Then we have*

$$\text{cost}_z(U, C) \in (1 \pm \epsilon) \text{cost}_z(V, C). \tag{11}$$

*Proof.* If $C_{\text{close}}^G \neq \emptyset$, by the property of uncolored group as in Lemma G.1, we know that $\forall x \in G, \text{dist}(x, C) \in (1 \pm \frac{\epsilon}{3z}) \cdot \text{dist}(x, c_i^*)$. So we have

$$\text{cost}_z(U, C) \in (1 \pm \epsilon) \text{cost}_z(U, c_i^*) \quad \text{and} \quad \text{cost}_z(V, C) \in (1 \pm \epsilon) \text{cost}_z(V, c_i^*).$$

By combining the above two inequalities and scaling $\epsilon$, we obtain (11).

In the other case, if $C_{\text{close}}^G = \emptyset$, Lemma G.1 implies $\forall x \in G, \text{dist}(x, C) > \frac{9z}{\epsilon} \cdot \text{dist}(x, c_i^*)$. By triangle inequality, we know that $\text{dist}(x, C) \in (1 \pm \frac{\epsilon}{3z}) \text{dist}(c_i^*, C)$. So we have,

$$\text{cost}_z(U, C) \in (1 \pm \epsilon) \cdot w_U(U) \cdot \text{cost}_z(c_i^*, C) \quad \text{and} \quad \text{cost}_z(V, C) \in (1 \pm \epsilon) \cdot w_V(V) \cdot \text{cost}_z(c_i^*, C),$$

moreover since $w_U(U) = w_V(V)$, we conclude (11) by scaling $\epsilon$. $\qquad\square$

We are ready to prove Lemma 3.7.

*Proof of Lemma 3.7.* It suffices to prove the following two directions separately.

$$\text{cost}_z^{(t)}(D, C) \leq (1 + \epsilon) \text{cost}_z^{(t)}(G_{\text{all}}, C) + \epsilon \cdot \text{cost}_z(P \setminus L^*, C^*), \tag{12}$$

$$\text{cost}_z^{(t)}(G_{\text{all}}, C) \leq (1 + \epsilon) \text{cost}_z^{(t)}(D, C) + \epsilon \cdot \text{cost}_z(P \setminus L^*, C^*), \tag{13}$$

and scale $\epsilon$. $\qquad\square$

**Proof of (12)** Let $(L, w_L)$ denote the outliers of $G_{\text{all}}$ with respect to $C$. Namely, $L \subset G, w_L(L) = t$ and

$$\text{cost}_z(G_{\text{all}} - L, C) = \text{cost}_z^{(t)}(G_{\text{all}}, C).$$

It suffices to find a weighted subset $(T, w_T)$ of $D$ such that $w_T(T) = t$ and

$$\text{cost}_z(D - T, C) \leq (1 + \epsilon) \text{cost}_z(G_{\text{all}} - L, C) + \epsilon \cdot \text{cost}_z(P \setminus L^*, C^*). \tag{14}$$

We define $T$ as the following. Recall that

$$G_{\text{all}} = \bigcup_{i \in [\beta k]} \bigcup_{G \in \mathcal{G}_i} G.$$

For every $G \in \mathcal{G}_i$, we add $\{p_{\text{close}}^G, p_{\text{far}}^G\}$ into $T$ and set

$$w_T(p_{\text{close}}^G) = \sum_{x \in L \cap G} \lambda_x, \quad w_T(p_{\text{far}}^G) = \sum_{x \in L \cap G} (1 - \lambda_x)$$

where we recall that $\lambda_x$ is the unique number in $[0, 1]$ such that $\text{dist}^z(x, c_i^*) = \lambda_x \cdot \text{dist}^z(p_{\text{close}}^G, c_i^*) + (1 - \lambda_x) \cdot \text{dist}^z(p_{\text{far}}^G, c_i^*)$.

If $G$ is an colored group, we apply Lemma G.3 to obtain

$$\text{cost}_z(D_G - (T \cap D_G), C) \leq (1 + \epsilon) \text{cost}_z(G - (L \cap G), C) + \epsilon \cdot \frac{\text{cost}_z(P_i, c_i^*)}{2k \log(z/\epsilon)} \tag{15}$$

Now suppose $G$ is an uncolored group, observe that by construction, $w_T(T \cap D_G) = w_L(L \cap G)$ and $\text{cost}_z(T \cap D_G, c_i^*) = \text{cost}_z(L \cap G, c_i^*)$. Applying Lemma G.4 in $D_G - (T \cap D_G)$ and $G - (L \cap G)$, we obtain that,

$$\text{cost}_z(D_G - (T \cap D_G, C) \leq (1 + \epsilon) \text{cost}_z(G - (L \cap G), C). \tag{16}$$

By Lemma G.1, there are at most $k \log(z/\epsilon)$ many colored groups in each cluster $P_i$, combining with (15) and (16), we have

$$\text{cost}_z(D - T, C)$$
$$= \sum_{i \in [\beta k]} \sum_{G \in \mathcal{G}_i} \text{cost}_z(D_G - (T \cap D_G), C)$$
$$\leq \sum_{i \in [\beta k]} \sum_{G \in \mathcal{G}_i} (1 + \epsilon) \text{cost}_z(G - (L \cap G), C) + k \log(z/\epsilon) \sum_{i \in [\beta k]} \epsilon \cdot \frac{\text{cost}_z(P_i, c_i^*)}{2k \log(z/\epsilon)}$$
$$\leq (1 + \epsilon) \text{cost}_z(G_{\text{all}} - L, C) + \epsilon \cdot \text{cost}_z(P \setminus L^*, C^*)$$

which is (14).

**Proof of (13)** Let $(T, w_T)$ denote the set of (total weight $w_T(T) = t$) outliers of $D$ with respect to $C$. Namely,

$$\text{cost}_z^{(t)}(D, C) = \text{cost}_z^{(t)}(D - T, C).$$

It suffices to find a weighted subset $(L, w_L)$ of $G$ such that $w_L(L) = t$ and

$$\text{cost}_z^{(t)}(G_{\text{all}} - L, C) \leq (1 + \epsilon) \cdot \text{cost}_z^{(t)}(D - T, C) + \epsilon \cdot \text{cost}_z(P \setminus L^*, C^*). \tag{17}$$

We construct $L$ as the following. For every $i \in [k]$, for every $G \in \mathcal{G}_i$, let $m_G = w_T(T \cap D_G)$ and let $(L_G, w_{(L_G)})$ denote a weighted subset of $G$ such that

$$\text{cost}_z^{(m_G)}(G, C) = \text{cost}_z^{(m_G)}(G - L_G, C).$$

In other words, $L_G$ is the subset of furthest $m_G$ weights of points to $C$ in $G$. Add $L_G$ into $L$ and set $w_L(x) = w_{(L_G)}(x)$ for every $x \in L_G$.

We prove $L$ satisfies (17). We do the following case study.

- If $G$ is a colored group, we simply apply Lemma G.3 to obtain

$$\text{cost}_z(G - L_G, C) \leq (1 + \epsilon) \text{cost}_z(D_G - (T \cap D_G), C) + \epsilon \cdot \frac{\text{cost}_z(P_i, c_i^*)}{2k \log(z/\epsilon)}. \tag{18}$$

- If $G$ is an uncolored group, and $C_{\text{close}}^G = \emptyset$, by Lemma G.4, we know that

$$\text{cost}_z(G - L_G, C) \leq (1 + \epsilon) \text{cost}_z(D_G - (T \cap D_G), C). \tag{19}$$

- If $G$ is an uncolored group, $C_{\text{close}}^G \neq \emptyset$, and $m_G \in \{0, |G|\}$, note that in this case $L_G = G$ or $L_G = \emptyset$. So we have

$$\text{cost}_z(G - L_G, c_i^*) = \text{cost}_z(D_G - (T \cap D_G), c_i^*) \tag{20}$$

  by the fact that $D_G$ is the two-point coreset of $G$, satisfying Definition 3.4. So in this case, the conditions of Lemma G.4 are satisfied. So we have,

$$\text{cost}_z(G - L_G, C) \leq (1 + \epsilon) \text{cost}_z(D_G - (T \cap D_G), C). \tag{21}$$

- If $G$ is an uncolored group, $C_{\text{close}}^G \neq \emptyset$, and $m_G \notin \{0, |G|\}$, we call such group a *special uncolored group* and prove in Lemma G.5 that there at most 2 special groups in every $\mathcal{G}_i$. (See Figure 5 for an illustration.) Then we use Lemma G.3 to obtain

$$\text{cost}_z(G - L_G, C) \leq (1 + \epsilon) \text{cost}_z(D_G - (T \cap D_G), C) + \epsilon \cdot \frac{\text{cost}_z(P_i, c_i^*)}{2k \log(z/\epsilon)}. \tag{22}$$

Combining (18), (19), (21), (22), and the fact that there are at most $k \log(z/\epsilon)$ colored groups and 2 special groups in each $\mathcal{G}_i$, we have

$$\text{cost}_z(G_{\text{all}} - L, C)$$
$$= \sum_{i \in [\beta k]} \sum_{G \in \mathcal{G}_i} \text{cost}_z(G - L_G, C)$$
$$\leq (1 + \epsilon) \sum_{i \in [\beta k]} \sum_{G \in \mathcal{G}_i} \text{cost}_z(D_G - (T \cap D_G), C) + (k \log(z/\epsilon) + 2) \cdot \sum_{i \in [\beta k]} \epsilon \cdot \frac{\text{cost}_z(P_i, c_i^*)}{2k \log(z/\epsilon)}$$
$$\leq (1 + \epsilon) \text{cost}_z(D - T, C) + \epsilon \cdot \text{cost}_z(P \setminus L^*, C^*).$$

**Lemma G.5.** *For a center set $C \subset \mathbb{R}^d$, $|C| = k$, in every $\mathcal{G}_i$, there are at most 2 special uncolored groups with respect to $C$.*

*Proof.* For the sake of contradiction, assume there are 3 special uncolored groups $G_1, G_2$, and $G_3$ in cluster $P_i$. Assume w.l.o.g. that $G_1$ is the furthest to center $c_i^*$ and $G_3$ is the closest one. Since $G_1$ is a special uncolored group, we know that $C_{\text{close}}^{G_1} \neq \emptyset$, so $\forall x \in G_1$, $\text{dist}(x, C) \in (1 \pm \epsilon) \cdot \text{dist}(x, c_i^*)$.

In particular, there exists an inlier $y_1 \in D_{G_1}$ such that $\text{dist}(y_1, C) \geq (1-\epsilon) \cdot \text{dist}(y_1, c_i^*)$. Similarly, there exists an outlier $y_3 \in G_3$ such that $\text{dist}(y_3, C) \leq (1+\epsilon) \cdot \text{dist}(y_3, c_i^*)$.

However, $G_1, G_2$ and $G_3$ are disjoint groups which are union of consecutive rings. So $\text{dist}(y_1, c_i^*) \geq 2\,\text{dist}(y_3, c_i^*)$ and this implies

$$
\begin{aligned}
\text{dist}(y_1, C) &\geq & (1-\epsilon) \cdot \text{dist}(y_1, c_i^*) \\
&\geq & 2(1-\epsilon) \cdot \text{dist}(y_3, c_i^*) \\
&> & (1+\epsilon) \cdot \text{dist}(y_3, c_i^*) \\
&> & \text{dist}(y_3, C)
\end{aligned}
$$

where we have used that $\epsilon < 0.3$. However, this contradicts to the fact that $y_1$ is an inlier but $y_3$ is an outlier. $\qquad\square$

## H    LOWER BOUNDS

We show in Theorem H.1 that the factor $m$ is necessary in the coreset size, even for the very simple case of $k = 1$ and one dimension, for $(k, m)$-ROBUST MEDIAN.

**Theorem H.1.** *For every integer $m \geq 1$, there exists a dataset $X \subset \mathbb{R}$ of $n \geq m$ points, such that for every $0 < \epsilon < 0.5$, any $\epsilon$-coreset for $(1, m)$-ROBUST MEDIAN must have size $\Omega(m)$.*

*Proof.* Fix $0 < \epsilon < 0.5$. Consider the following instance $X = \{x_0, \ldots, x_m\} \subset \mathbb{R}^1$ of size $n = m + 1$, where $x_0 = 0$ and $x_i = i$ for $i \in [m]$. Suppose $(S, w_S)$ is an $\epsilon$-coreset for $(k, m)$-ROBUST MEDIAN.

We first claim that $w_S(S) \geq m + 1 - \epsilon$. This can be verified by letting center $c \to +\infty$, and we have

$$
\frac{\text{cost}_1^{(m)}(S, c)}{\text{cost}_1^{(m)}(X, c)} = w_S(S) - m \in 1 \pm \epsilon.
$$

Next, let $c = \frac{x_{i-1} + x_i}{2}$ for some $i \in [m+1]$, which implies that $\text{cost}_1^{(m)}(X, c) = |x_i - c| = 0.5$, i.e., the distance to the nearest-neighbor of $c$ in $X$. Suppose both $x_{i-1}$ and $x_i$ are not in $S$ and we have

$$
\begin{aligned}
\text{cost}_1^{(m)}(S, c) &\geq & (w_S(S) - m) \cdot \min_{x \in S} d(x, c) & \\
&\geq & (1-\epsilon) \cdot \min_{x \in S} d(x, c) & (w_S(S) \geq m + 0.6) \\
&\geq & (1-\epsilon) \cdot 1.5 & (x_{i-1}, x_i \notin S) \\
&> & 0.75 & (\epsilon < 0.5) \\
&> & (1+\epsilon) \cdot \text{cost}_1^{(m)}(X, c), & (\epsilon < 0.5)
\end{aligned}
$$

which is a contradiction. Hence, either $x_{i-1}$ or $x_i$ must be contained in $S$. It is not hard to conclude that $|S| \geq \frac{m-1}{2}$, which completes the proof. $\qquad\square$

