# OpenReview forum: "Near-optimal Coresets for Robust Clustering"
_ICLR.cc/2023/Conference — ICLR 2023 notable top 5%_

### Official Review · Reviewer_qzTd · 2022-10-24

**Confidence:** 3
**Correctness:** 4
**Technical Novelty And Significance:** 3
**Empirical Novelty And Significance:** 3
**Recommendation:** 8

**Clarity, Quality, Novelty And Reproducibility:**

The paper is very well written, including a good explaination of "The Power of Uniform Sampling", whose techniques they heavily rely on foe their proble. Both quality and novelty are at par.


**Strength And Weaknesses:**

strength:
- the result improves the coreset size by expotential factor in k abd m, by bringing it down to nearly optimal (m + ply(n,eps^(-1))).
- it presents extensive experimental results on real world datasets.

weakness/suggestions:
- a comparative discussion about its coreset vs exponential size coreset will be useful to motivate towards the probem
- are the groups consist of rings with at most 1 point? The discussion about groups could be further improved.
- a proof sketch of lemma 3.6 and 3.7 in the main paper could be useful.

**Summary Of The Paper:**

The paper shows constructs coreset for (k,z,m)-robust clustering problem, where k is the number of clusters over the distance measure between two points are || - ||_2^z and m is the number of outlier points. The coreset size linearly depends on m, poly(k,eps^(-1)) and exponentially depends on z. The running time of the algorithm is yime taken to get a constant factor aprooximation of the problem plus O(nkd).

**Summary Of The Review:**

The paper tackles a very important problem called robust clustering, which has many common applications. It improves the coreset size by exponential factor. It also shows emperical results on real datasets to back its theoretical claims and to show its benifits in improving scalability.

---

> ### Author Response · Authors · 2022-11-14
> **Responses to Reviewer qzTd**
>
> We thank the reviewer for the positive and insightful comments! We respond in detail to your specific concerns in the following.
>
> >''a comparative discussion about its coreset vs exponential size coreset will be useful to motivate towards the problem''
>
> The previous exponential-size coreset (Feldman and Schulman, 2012) has a $(k + m)^{k + m}$ size factor. This exponential dependence on $k$ and $m$ is likely to be impractical, since typical values of $k$ and/or $m$ may be $O(\log n)$. In addition, as observed in our experiments, the value of $m$ can be as large as $1500$ in real datasets, so the dependence $(k + m)^{k +m}$ in (Feldman and Schulman, 2012) is prohibitively large. The inferior performance of the exponential-size coreset is also observed in our experiments (see Section 4.1).
>
> We will add a brief discussion about this in our next version.
>
>
> >''are the groups consist of rings with at most 1 point? The discussion about groups could be further improved.''
>
> We identify rings in a similar way as in (Braverman et al., 2022), where the rings are formed greedily based on some function of cost (which does not directly depend on the number of points inside). Hence, we may include into a group some ring with only one point.
>
> >''a proof sketch of Lemmas 3.6 and 3.7 in the main paper could be useful.''
>
> We will provide a proof sketch of Lemmas 3.6 and 3.7 in the main text of the next version. A highlight to how the outliers affect the arguments can be found in thread: https://openreview.net/forum?id=Nc1ZkRW8Vde&noteId=jbNKcffdzn9

---

### Official Review · Reviewer_7BQh · 2022-10-24

**Confidence:** 4
**Clarity, Quality, Novelty And Reproducibility:** Paper is written clearly and it is ea…
**Correctness:** 4
**Technical Novelty And Significance:** 4
**Empirical Novelty And Significance:** 3
**Recommendation:** 8

**Strength And Weaknesses:**

S1: Strong results and significant improvement of the state of the art.
S2: Timely contribution as the non-robust version has been only recently settled and having robust algorithms is a hot topic.
S3: Great practical performance despite the large dependence on k and epsilon.

W1: There is still some gap between upper and lower bounds.

**Summary Of The Paper:**

This paper presents near-optimal coresets for robust Euclidean clustering. Given k, z, m, and a set of n points P in R^d, we look for a k-clustering of P excluding a set of outliers L of size at most m, minimizing the sum of (power z of) distances of points to their centers. This gives k-median and k-means objectives for z = 1, 2.

The authors present a coreset of size O(m + poly(k/epsilon)), which adds error epsilon to the approximation.
- Best coresets for non-robust problem have size poly(k / epsilon).
- Linear dependence on m is necessary (the paper gives a simple example).
- Dependence of k^3 and epsilon^5 (k-median) or epsilon^8 (k-means) sound too big, but the experiments show that one can use much smaller dependence and beat prior work.
- Omega(k/epsilon^2) is a lower bound for the non-robust case.
- Prior work for robust algorithms either had exponential dependence on m or bicriteria for the number of outliers.

This work heavily builds on the techniques from the recent Braverman et al. [FOCS 2022], though certain pieces need to be adapted to the outlier setting.

**Summary Of The Review:**

Significant improvement of the state of the art for an important problem, and very good practical performance.

---

> ### Author Response · Authors · 2022-11-14
> **Responses to Reviewer 7BQh**
>
> We thank the reviewer for the positive and insightful comments!
>
> For the weakness that you mention:
>
> >''There is still some gap between upper and lower bounds.''
>
> Indeed, the main open question is to tighten the dependence of $k$ and $\varepsilon$ (as the dependence in $m$ is already tight). However, this probably requires first improving the framework of (Braverman et al., 2022) for non-robust clustering, which already has a somewhat high dependence on $k$ and $\varepsilon$.

---

### Official Review · Reviewer_1Ghq · 2022-10-25

**Confidence:** 3
**Correctness:** 4
**Technical Novelty And Significance:** 4
**Empirical Novelty And Significance:** 4
**Recommendation:** 8

**Clarity, Quality, Novelty And Reproducibility:**

- I believe this paper significantly contributes on the field of algorithms for robust ML.

- The result looks correct and the writing is intuitive and clear.

- The authors provide the code and data for their experiments.

**Strength And Weaknesses:**

Strengths:
- The robust clustering problem is important in practice. Coresets are very relevant in large scale and dynamic applications where going maintaining and processing all the data points is impractical.
- The algorithm is simple and intuitive.
- The experimental results on four public datasets are promising. In particular, the authors' coresets have significantly better quality than trivial ones, e.g. uniform sampling. In addition, the coreset generation time seems to be >10x faster than the time if one were to run robust clustering without a coreset. This shows that coreset generation can be used as a preprocessing step to significantly speed up robust clustering.

Weaknesses:
- I would like to have seen a comparison with the algorithm of Bhaskara et al. In theory it gives a $O(1)$-approximation, but how does it do in practice compared to the coreset approach? Also, there are other, simpler heuristics for coreset generation that would be nice to be considered. For example, what happens if I just perform uniform sampling on the clusters of Bhaskara (plus the outliers)?
- There is some confusion with symbols in various places: $P$ vs $X$, $n$ not defined, what is $c^*$ in Lemma F.1, etc.

**Summary Of The Paper:**

This paper studies coresets for robust clustering. A coreset is a small proxy of the dataset, which is yet sufficient to perform the task at hand while not significantly compromising the quality of the solution. The goal of the robust clustering task is to cluster a set of data points that contains a number of adversarial outliers. Concretely, given a clustering objective $cost$ (e.g. $k$-means) and a set of points $P$, the robust clustering objective with $m$ outliers is defined as the minimum cost achievable by a clustering that is allowed to remove up to $m$ datapoints: $\min_{\text{clustering }C} \min_{M\subseteq P, |M|\leq m} cost(P \backslash M, C)$.

The main result of this paper is a coreset for the robust clustering problem with a range of clustering cost objectives that includes $k$-means and $k$-median. Specifically, the coreset has size $O(m) + \mathrm{poly}(k \epsilon^{-1})$, where $k$ is the desired number of clusters and $\epsilon$ is an error tolerance. This is quite satsifying since previous work shows that $\mathrm{poly}(k \epsilon^{-1})$ is tight for the non-robust case (even though the polynomial dependency doesn't quite match the lower bound). The runtime is O(|P| k d), where d is the dimension of the data.

The idea is to first run an approximate but fast robust clustering algorithm from previous work (Bhaskara et al), and then define the coreset as the set of worst fitted points wrt this clustering plus a random sample. The most technical part is this sampling process, which builds on previous work of Braverman et al.

**Summary Of The Review:**

In summary, I find this paper interesting and I think it makes significant theoretical progress in a fundamental ML problem, and is also promising for practical applications. I recommend acceptance.

---

> ### Author Response · Authors · 2022-11-14
> **Responses to Reviewer 1Ghq**
>
> We thank the reviewer for the positive and insightful comments! We respond in detail to your specific concerns in the following.
>
> >''I would like to have seen a comparison with the algorithm of Bhaskara et al. In theory it gives a $O(1)$-approximation, but how does it do in practice compared to the coreset approach?''
>
> In fact, Bhaskara et al. is an approximation algorithm, while ours is a coreset, so they are not directly comparable.
>
> Nonetheless, we did show how our coresets speed up the Bhaskara et al. algorithm in Section 4.2. Specifically, we run on top of our coreset the Bhaskara et al. algorithm followed by a Lloyd heuristic (which we refer to LL in the paper). We observe a significant speedup of LL, compared with running it directly without the coreset, while still achieving a similar clustering cost.
>
> >''Also, there are other, simpler heuristics for coreset generation that would be nice to be considered. For example, what happens if I just perform uniform sampling on the clusters of Bhaskara (plus the outliers)?''
>
> Actually, the approach that you suggested may be interpreted as a more general one: find an initial solution using some approximation algorithm, add the outliers to the coreset, and do uniform sampling on each initial cluster to form the coreset. We evaluated this general approach, and we find it performs nearly the same as our coreset on various datasets.
>
> This is somewhat surprising, but we find it likely due to the special structure of the data. Indeed, we observe that, for all these datasets, our algorithm finds very few rings (2 or 3) in each initial cluster, each ring consists of roughly the same number of points, and the number of points in groups is very small. Hence, the overall behavior of our algorithm is roughly a uniform sampling on each initial cluster, which mimics the new baseline you suggested.
>
>
> >''There is some confusion with symbols in various places: $P$ vs $X$, $n$ not defined, what is $c^∗$ in Lemma F.1, etc.''
>
> Thanks for pointing these out, and we will make another pass to make sure the notations are properly defined and consistently used.
>
> For the specific question about $c^*$: $c^*_i$ is the $i$-th center of $C^*$, where $C^*$ is an $(\alpha, \beta, \gamma)$-approximation defined in Definition 2.2.

---

### Official Review · Reviewer_A4Ec · 2022-10-25

**Confidence:** 4
**Correctness:** 4
**Technical Novelty And Significance:** 3
**Empirical Novelty And Significance:** 2
**Recommendation:** 8

**Clarity, Quality, Novelty And Reproducibility:**

I found it difficult from the paper itself to understand which rings are classified as groups and which are not. I had to read the Braverman paper to understand. I think the notion of marked (heavy) rings with higher costs and unmarked ones is not clearly stated here. It would be useful if it is explained clearly in the paper.

A detailed time analysis of the algorithm will be useful. Also, it will be good to give an overview of the computational complexity of original by problem.

The size of L in proof of theorem 3.1 is $t$ or $m$? Also, can you please explain how does the proof work for all $C$ and not only for a fixed $C$. In other words, I am a bit unclear as the outliers change with change in centers, so an explanation as to why same subset works as the coreset will be appreciated.

Usually in coreset literature while performing experiments you solve the problem on the full data, take the cost as 'optimal' cost and also solve on coreset and plug the solution obtained from coreset back in full data and compare that cost obtained with the 'optimal' cost. I understand that what you report here is the coreset guarantee with random set of centers, however the above measure might be useful as well.

Also, if we use the coreset obtained using Braverman et al. technique for vanilla clustering as a baseline how different it will be compared to uniform or sensitivity-based sampling. Can you provide your comments?

I did not check proofs in the appendix thoroughly, but they appear ok. Code is provided and I believe the empirical results are reproducible.

**Strength And Weaknesses:**

Strengths:
1. The problem of robust clustering is well studied and important one while at the same time being computationally expensive. Using coresets for such computationally expensive problems is a justified and popular approach.  This paper improves the size of coreset for the problem significantly and also provides a lower bound for the same.
2. A good set of experiments is performed to validate theoretical claims and the code is provided.

Weaknesses:
1. The paper appears to rely heavily on the paper by Braverman et al. 2022 for the coreset construction idea and the proof techniques. In that regard seems to be lacking in novelty. The novelty section needs to be highlighted better as to what are the technical challenges faced in comparison to the Braverman et al. paper. Do mention clearly the subtleties when adopting the technique to incorporate outliers.
2. While the paper is written nicely in terms of language and notations, there are a few points which I believe will enhance the completeness aspect. I list them below in the section on clarity.

**Summary Of The Paper:**

This paper presents coresets for a robust version of the $k$-clustering problem where the cost of clustering is taken over the dataset after removing $m$ outliers. The coreset size is polynomial in $k$ the number of clusters and linear in $m$, the number of outliers. The paper also shows a lower bound on the coreset size implying that the linear dependence on $m$ is necessary. The theoretical results are validated with experiments on real-world datasets.

**Summary Of The Review:**

Overall, I believe that the results of this paper will be of interest to the community. However, the paper needs to highlight its contributions in terms of novelty more clearly and also improve the writing to include few concepts for the sake of completeness. I believe if these changes are made the paper will present a better case for acceptance.

---

> ### Author Response · Authors · 2022-11-14
> **Responses to Reviewer A4Ec - Technical Novelty**
>
> We thank the reviewer for the very comprehensive and insightful comments! We start with addressing a major concern about technical novelty. We will address other comments in the second response.
>
> #### Technical challenge and novelty
>
> >''The paper appears to rely heavily on the paper by Braverman et al. 2022 for the coreset construction idea and the proof techniques. In that regard seems to be lacking in novelty. The novelty section needs to be highlighted better as to what the technical challenges faced in comparison to the Braverman et al. paper. Do mention clearly the subtleties when adopting the technique to incorporate outliers.
> Also, can you please explain how the proof works for all $C$ and not only for a fixed $C$. In other words, I am a bit unclear as the outliers change with change in centers, so an explanation as to why the same subset works as the coreset will be appreciated.''
>
>
> The presence of outliers can affect the entire argument from (Braverman et al., 2022) in many ways, and we highlight them in the following.
>
> Recall that, the overall framework decomposes the space/input into collections of *rings* and *groups*, and one applies uniform sampling for rings, and two-point coresets for groups. Let's consider a fixed center set $C$, and discuss how the outliers complicate the ring and group part of the analysis, respectively. We focus on the $z = 1$ case.
>
>
> * **Rings (Lemma 3.6)**. Consider some ring $R = \mathrm{ring}(c^*_i, r, 2r)$. Similar to (Braverman et al., 2022), we construct a coreset for $R$ simply by uniform sampling. In (Braverman et al., 2022), the error incurred by this uniform sampling was bounded by $\varepsilon \cdot \mathrm{cost}(R, C)$ which is $\varepsilon$ times the total cost *without* outliers from $R$ to $C$ (ignoring a "neglectable" additive term $O(\varepsilon |R| r)$ which can be directly charged to OPT). However, the mentioned error term $\varepsilon \cdot \mathrm{cost}(R, C)$ cannot be charged to the objective $\mathrm{cost}^{(m_R)}(R, C)$ when outliers are present, where $m_R$ is the number of outliers in $R$ (with respect to the $C$ that we fixed). This is because $\mathrm{cost}^{(m_R)}(R, C)$ can be very small, or even close/equal to $0$, when $m_R$ is large (e.g., $m_R \approx |R|$). Therefore, in Lemma E.4, we give an alternative upper bound when $m_R$ is large (and use a similar argument as before when $m_R$ is small), which eventually charges the cost of $R$ to the OPT.
> * **Groups (Lemma 3.7)**. Consider some group $G$. We use a similar two-point coreset for $G$ as in (Braverman et al., 2022). Let's call a group *bad* if the error incurred by the two-point coreset is much larger than $\varepsilon\cdot \mathrm{cost}^{(m_G)}(G,C)$, which is $\varepsilon$ times the contribution of $G.$ We focus our discussion on bounding the error of bad groups. We first show even with outliers (in Lemma F.1), the error for each bad group is at most $\varepsilon$ times the contribution of $G$ in OPT. Hence, it remains to bound the number of bad groups. In (Braverman et al., 2022), only a colored group can be bad, and the number of them is bounded by $O(k \log \frac{z}{\varepsilon})$. However, in the outlier setting a key difference to (Braverman et al., 2022) is that, uncolored groups can also be bad (due to a reason similar to that for rings: $\mathrm{cost}^{(m_G)}(G, C)$ may be too small when the number of outliers $m_G$ in $G$ is large), and we call them *special uncolored groups*.  To bound the number of them, in Lemma F.5 we make a crucial geometric observation: even though special uncolored groups may significantly change for varying $C$'s, an invariant is that they must always be consecutive uncolored groups, which implies the number of such groups is $O(k)$.
>
> Finally, in addition to the above new steps, it is also necessary to make these bounds for rings/groups work for all numbers of outliers $0 \leq t \leq m$ *simultaneously*, since one does not know in advance how many outliers reside each ring/group due to the arbitrarily chosen $C$.
>
>
> We will incorporate part of this discussion into our introduction.

---

> > ### Author Response · Authors · 2022-11-14
> > **Responses to Reviewer A4Ec - Other Issues**
> >
> > ### Responses to other comments
> >
> > >''I found it difficult from the paper itself to understand which rings are classified as groups and which are not. I had to read the Braverman paper to understand. I think the notion of marked (heavy) rings with higher costs and unmarked ones is not clearly stated here. It would be useful if it is explained clearly in the paper.''
> >
> >
> > In the (Braverman et al., 2022) paper, they mark some of the rings (which they call heavy rings), then group the remaining (unmarked) rings into groups. Our notion of "ring" corresponds to their "marked ring", and our "group" is the same as theirs. However, we do not need the concept of unmarked rings explicitly, since we only need to deal with the groups that are formed from them (and the construction of groups follows from a black box in Braverman et al). We will add a remark in our revision to discuss this in detail.
> >
> >
> > >''A detailed time analysis of the algorithm will be useful. Also, it will be good to give an overview of the computational complexity of original by problem.''
> >
> > We will add a detailed analysis of the time complexity in our proof. Specifically, the running time of Algorithm 1 is dominated by Lines 3 and 4, both taking $O(nkd)$ time.
> >
> > Apart from the steps of building the coresets, the time complexity for the initial tri-criteria approximation was discussed in the first paragraph of Section 1.3 and Section A.
> >
> > >''The size of $L$ in proof of theorem 3.1 is $t$ or $m$?''
> >
> > The size of $L$ is $t$, and we will add a clarification to the text to make it more clear.
> >
> > >''Usually in coreset literature while performing experiments you solve the problem on the full data, take the cost as 'optimal' cost and also solve on coreset and plug the solution obtained from coreset back in full data and compare that cost obtained with the 'optimal' cost. I understand that what you report here is the coreset guarantee with random set of centers, however the above measure might be useful as well. ''
> >
> > We followed your suggestion and tried to conduct an extra experiment to test the size-error tradeoff, with respect to the (near-)optimal solution. Since computing the optimal solution is NP-hard, we use a local-search algorithm instead.
> >
> > Interestingly, the results show that all coresets, including ours and all baselines that we consider (such as uniform sampling), perform equally well under this error measure. In particular, we observe a $<5\%$ error is achieved by $\approx m + 300$ points.
> >
> > This could mean only preserving the (near-)optimal, which is sometimes called *weak* coreset, is much easier than preserving all centers, as our *strong* coreset definition requires. It is an interesting question to explore the construction/bounds for the weak coresets.
> >
> > >''Also, if we use the coreset obtained using Braverman et al. technique for vanilla clustering as a baseline how different it will be compared to uniform or sensitivity-based sampling. Can you provide your comments?''
> >
> > We implemented your suggested baseline, and it turns out that the Braverman et al. coreset, which is designed only for vanilla clustering, has a very close performance to our new coreset overall, and it outperforms sensitivity-sampling and uniform sampling.
> >
> > This could mean that for the particular real datasets which we use, the outliers may not have a significant effect on the cost function. However, this cannot be interpreted as the additional steps in our coreset construction not necessary in general, since we already show an $\Omega(m)$ lower bound in Theorem 1.2 for the coreset size.

---

> ### Comment · Reviewer_A4Ec · 2022-11-24
> **After Rebuttal**
>
> I have read the other reviews and also the author rebuttal and I have raised my score. I request the authors to add the clarifications made in the rebuttal to the final version of the paper

---

> > ### Author Response · Authors · 2022-11-24
> > **Reply to the reviewer**
> >
> > Thanks a lot for increasing your support! We will add all clarifications to the final version as suggested.

---

### Decision · Program_Chairs · 2023-01-20

**Decision:**

Accept: notable-top-5%

**Justification For Why Not Higher Score:**

N/A

**Justification For Why Not Lower Score:**

see summary.

**Metareview: Summary, Strengths And Weaknesses:**

Strong results on an important problem with convincing experiments, everyone thought it was a clear accept.

**Note From Pc:**

if the above contains the word "oral" or "spotlight" please see: "oral" presentation means -> notable-top-5% and "spotlight" means -> notable-top-25%. As stated in our emails, we are disassociating presentation type from AC recommendations

**Summary Of Ac-Reviewer Meeting:**

N/A